# Estimating the intrinsic dimensionality using Normalizing Flows

**Christian Horvat**
Department of Physiology
University of Bern
christian.horvat@unibe.ch

**Jean-Pascal Pfister**
Department of Physiology
Bern, Switzerland
jeanpascal.pfister@unibe.ch

## Abstract

How many degrees of freedom are there in a dataset consisting of $M$ samples embedded in $\mathbb{R}^D$? This number, formally known as *intrinsic dimensionality*, can be estimated using nearest neighbor statistics. However, nearest neighbor statistics do not scale to large datasets as their complexity scales quadratically in $M$, $\mathcal{O}(M^2)$. Additionally, methods based on nearest neighbor statistics perform poorly on datasets embedded in high dimensions where $D \gg 1$. In this paper, we propose a novel method to estimate the intrinsic dimensionality using Normalizing Flows that scale to large datasets and high dimensions. The method is based on some simple back-of-the-envelope calculations predicting how the singular values of the flow's Jacobian change when inflating the dataset with different noise magnitudes. Singular values associated with directions normal to the manifold evolve differently than singular values associated with directions tangent to the manifold. We test our method on various datasets, including 64x64 RGB images, where we achieve state-of-the-art results.

## 1   Introduction

Learning low-dimensional representations of high-dimensional data is becoming increasingly important in the era of big data. Therefore, representation learning is a very active area of research [33] with a wide range of applications ranging from neuroscience [27], molecular biology [28], bioinformatics [12] or image analysis [21]. But how many low-dimensional variables are exactly needed to accurately describe the original data? Unfortunely, this number is typically not known and needs to be estimated - a task formally known as *intrinsic dimensionality* (ID) estimation.

Moreover, estimating the ID turned out to be an usefull tool for understanding the functioning of Neural Networks and generative models [23, 11, 1, 31]. In a recent study, [29] verified that the ID plays a large role in deep learning on natural images and hence motivated the necessity to better estimate the ID in this setting. In the context of generative models, the ID corresponds to the number of latent variables generating the data. Some of these generative models do even rely on knowing the exact number of latent variables generating the data which limits their applicability to real-world problems where this number is unknown [3, 16].

ID estimation is mathematically challenging. Most approaches rely on the assumption that, locally, the samples are uniformly distributed, and therefore the ID can be estimated using nearest neighbor statistics [4]. However, such nearest neighbor methods do not scale to large datasets consisting of $M \gg 1$ samples as the complexity scales with $\mathcal{O}(DM^2)$ where $D$ is the dimensionality of the embedding space and accounts for calculating the (Euclidean) distance [1] . Thus, this scaling issue is amplified whenever the embedding space $D$ is of the order of the sample size $M$, $D = \mathcal{O}(M)$.

---

[1]Therefore, some line of research tries to reduce the complexity of finding the nearest neighbor, see e.g. [25].

36th Conference on Neural Information Processing Systems (NeurIPS 2022).

Additionally, nearest neighbor methods suffer from the curse of dimensionality in the sense that they perform badly in high dimensions $D \gg 1$ [5, 4, 32]. In this paper, we propose to overcome these scalability issues by estimating the ID using standard NFs which do scale to large datasets and dimensions.

Our high-level idea is very simple and based on some back-of-the-envelope calculations which predict how the flow's eigenvalues change depending on the amount of noise injected into the data prior to training. For directions of small variability (i.e. off-manifold directions), the corresponding eigenvalues should decrease at a rate predicted by our theory. However, for directions of great variability (i.e. manifold directions), the corresponding eigenvalues will behave differently which allows us to estimate $d$. We show the soundness of our approach on various datasets, including RGB images of resolution $64 \times 64$.

## 2  Problem statement, background and notations

Here we briefly discuss the problem of estimating the ID, introduce the concept of an NF and additional notations which will be used throughout the paper.

**Estimating ID:** An $d-$dimensional manifold $\mathcal{M}$ is a set of points that are locally diffeomorphic to subsets of $\mathbb{R}^d$, see e.g. [22] for a formal definition of manifolds. Given $M$ samples $x_1, \ldots, x_M$ from such a $d-$dimensional manifold embedded in $\mathbb{R}^D$, $d < D$, the task is to estimate $d$. More generally, if the dataset is living on unions of manifolds with different IDs, the task is to estimate $d$ locally for a given point $x^*$ [2] . For real-world datasets, however, samples are corrupted by noise making the dataset full-dimensional, i.e. $d = D$. In this case, estimating the ID corresponds to estimating how many main degrees of freedom there are in the observed dataset. We refer to those degrees of freedom to directions of large variability in the following.

**Standard Normalizing Flows:** In the simplest case, an NF $f_\theta$ is a change of variable transforming a sample from the unknown distribution $p(x)$ to a sample from a standard Normal $\mathcal{N}(0, I_D)$. If $\mathcal{M}$ is full-dimensional, i.e. $d = D$, then the resulting density is given by $p_\theta(x) = |\det J_{f_\theta^{-1}}(u)|^{-1} \mathcal{N}(u; 0, I_D)$ where $u = f_\theta^{-1}(x)$, and the parameters $\theta$ are updated to minimize

$$D_{KL}(p(x)||p_\theta(x)) = -\mathbb{E}_{x \sim p(x)}[\log p_\theta(x)] + const. \tag{1}$$

For the general case where $d < D$, the general change of variable formula needs to be considered, see e.g. [17, 8, 10, 3]. From now on we omit the NF's dependence on the parameter $\theta$ and write $f$ instead of $f_\theta$.

**Notations:** We denote the data density with support on $\mathcal{M}$ as $p(x)$. Given a sample $x \sim p(x)$, if we add some Gaussian noise $\varepsilon$ to it, the resulting random variable $\tilde{x} = x + \varepsilon$ has the following density

$$q_{\sigma^2}(\tilde{x}) = \int_{\mathcal{M}} \mathcal{N}(\tilde{x}; x, \sigma^2 I_D) p(x) dx \tag{2}$$

where $I_D$ is the $D \times D$ unit matrix.
The Jacobian of a flow $f : \mathbb{R}^D \to \mathbb{R}^D$ has $D$ eigenvalues and singular values, which we both denote by $\lambda^{(1)}, \ldots, \lambda^{(D)}$.

## 3  Related work

Estimating the ID is a well-studied problem with a vast literature, see [5, 4] for exhaustive overviews.

**Nearest Neighbor methods:** Historically, two main branches have developed: global and local methods [4]. Global methods estimate the dimensionality of the dataset globally with methods based on PCA is the most famous representative of this class, while local methods do estimate it locally. Consistent local estimators are based on nearest neighbor statistics. Intuitively, the volume of a $d-$dimensional ball scales as $r^d$ with its radius $r$, and, therefore, the number of nearest neighbors within the $r-$ball of a point should scale similarly. This intuition was formalized by [13] and generalized in [14]. Recently, [9] derived the distribution of the distance to the second nearest neighbor - a power law with the ID as the exponent. We choose this method, coined twoNN, as a

---

[2]Note, that a local estimator can be used as a global ID estimator by simply averaging over different samples.

representative for nearest neighbor methods and compare its performance with our method in Section 5 .

**Methods based on NN:** Neural Networks (NN) scale to large datasets. Surprisingly, to the best of our knowledge, next to our method, there exists only one method which attempts to *directly* estimate the ID using an NN based method [32], coined LIDL (see below for a detailed description). However, there exist different *indirect* methods which use performance metrics to estimate the ID. The general idea is to train different models with different latent dimensionalities and compare their performance on a chosen metric, see e.g. [2, 34, 3]. Arguably, the model yielding the best result on this metric corresponds to the right latent dimensionality. However, the estimate will be relative to the chosen performance metric and therefore might not reflect the true ID. As an example of this relativity, we refer to Figure 14 in [3] which use their NF-based method to estimate the ID of the CelebA dataset [24]. They get different results when using the FID-score [15] or reconstruction error as performance metric. In [30], it was argued that the Jacobian of the encoder $f : \mathbb{R}^D \to \mathbb{R}^d$ of an Autoencoder can give information of directions of contraction. Intuitively, manifold directions correspond to directions of low contraction while off-manifold directions correspond to directions of high contraction. The gist of this idea is very similar to our intuition which we will present Section 4 as directions of contraction can be directly related to the singular values of the encoder's Jacobian. However, the number of contractive directions will only give a lower bound on the ID.

**LIDL:** [32] propose to use NFs to estimate the ID. LIDL stands for locally intrinsic dimensionality likelihood and is based on the assumption that the inflated log-likelihood $\log q_{\sigma^2}(x)$ scales linearly in $\sigma^2$ with the normal space dimension as slope. More precisely, they argue that the inflated distribution $q_{\sigma^2}(x)$ can be written as the product of $p(x)$ and the noise distribution restricted on the normal space, i.e. $q_{\sigma^2}(x) = p(x)\mathcal{N}(x; x, I_{D-d})$ where $\mathcal{N}(x; x, I_{D-d})$ is a $(D - d)-$dimensional Gaussian in the normal space of $x$. This intuition is mathematically confirmed in [17] under certain conditions on the manifold and generating latent distribution. Thus, [32] estimate the rate at which the log-likelihoods change depending on the noise magnitude $\sigma^2$, and therefore train different models with different noise magnitudes $\bar{\sigma}_1^2 < \cdots < \bar{\sigma}_N^2$ to estimate that change.

LIDL heavily relies on the assumption that $q_{\sigma^2}(x) = p(x)\mathcal{N}(x; x, I_{D-d})$, which is generally only true for sufficiently smooth manifolds and latent distributions, see [17] for details. Even if these conditions are fulfilled, the outcome of LIDL depends on the noise magnitudes $\bar{\sigma}_1^2 < \cdots < \bar{\sigma}_N^2$ used to estimate the log-likelihood rate. However, the optimal range again depends on both the manifold and generating latent distribution which is hard to estimate a priori [17]. We will revisit these issues in the experiments, Section 5.

## 4   Method

How can we estimate the intrinsic dimensionality of data sampled from a manifold using an NF?

**Simplest scenario:** Assume the simple case where the target distribution $p(x)$ is a normal distribution, i.e. $p(x) = \mathcal{N}(\mu, \Sigma)$ with $\mu \in \mathbb{R}^D$ and

$$\Sigma = \mathrm{diag}(\sigma_1^2, \ldots, \sigma_D^2), \quad \sigma_i^2 > 0. \tag{3}$$

Thus, we assume that the data manifold is given by the entire embedding space, $\mathcal{M} = \mathbb{R}^D$. Note, however, that if $\sigma_i^2 \to 0$, $i = 1, \ldots, D - d$, this corresponds to samples from a $d-$dimensional manifold embedded in $\mathbb{R}^D$. Then, the true NF $f$ transforming $x \sim \mathcal{N}(\mu, \Sigma)$ into $u = f^{-1}(x)$ distributed according to $\mathcal{N}(0, I_D)$ is given by $f_i^{-1}(x) = (x_i - \mu_i)/\sigma_i, i = 1, \ldots, D$, with Jacobian

$$J_{f^{-1}}(x) = \mathrm{diag}\left(\frac{1}{\sigma_1}, \ldots, \frac{1}{\sigma_D}\right). \tag{4}$$

Therefore, an eigenvector of $J_{f^{-1}}(x)$ corresponding to a large eigenvalue $\lambda^{(i)}$ (i.e. small $\sigma_i$ ) is in direction of small variability in the data. It is easy to see that the same holds for more general covariance matrices $\Sigma$. The true NF $f$ transforming $x \sim \mathcal{N}(\mu, \Sigma)$ into $u = f^{-1}(x)$ distributed according to $\mathcal{N}(0, I_D)$ is given by

$$f^{-1}(x) = \Sigma^{-\frac{1}{2}}(x - \mu) \tag{5}$$

with Jacobian

$$J_{f^{-1}}(x) = \Sigma^{-\frac{1}{2}} = S^T D^{-\frac{1}{2}} S \tag{6}$$

where $\Sigma = S^T D S$, $D = \text{diag}(\sigma_1^2, \ldots, \sigma_D^2)$ and $S$ is orthonormal and consists of the eigenvectors of $\Sigma$. Thus, the eigenvalues of $J_{f^{-1}}(x)$ are given by $\frac{1}{\sigma_1}, \ldots, \frac{1}{\sigma_D}$ as for the simplest scenario.

**From simple to the generic scenario:** In general, the NF $f^{-1}$ non-linearly transforms a sample $x \sim p(x)$ such that $u = f^{-1}(x)$ is distributed according to $\mathcal{N}(0, I_D)$. Hence, Equation (5) does not hold anymore. However, Equation (5) is a good first-order approximation as locally the flow acts linearly,

$$f^{-1}(x) \approx f^{-1}(x^*) + J_{f^{-1}}(x^*)(x - x^*). \tag{7}$$

What follows is that a flow $f^{-1}$ not only transforms globally samples from $p(x)$ to samples from a Gaussian, but also locally. This was already observed in [7] (Lemma 1) for $f$, i.e. for the direction from latent to data space. The same holds true for $f^{-1}$ which immediately follows from the delta method [26] .

**Lemma 1** *Let $x^*$ be a sample from $p(x)$, and let $f^{-1}$ be the flow such that $z^* \sim \mathcal{N}(0, I_D)$. where $z^* = f^{-1}(x^*)$. Let $\varepsilon \sim \mathcal{N}(0, I_D)$ and $s > 0$ be a scalar. Consider samples generated by $\tilde{z} = f^{-1}(x^* + s\varepsilon)$. Then, for $s \to 0$ we have that $\frac{1}{s}(\tilde{z} - z^*) \longrightarrow \mathcal{N}(0, J_{f^{-1}}(x^*)^T J_{f^{-1}}(x^*))$ where the convergence is in distribution.*

Lemma 1 states that a small ball around $x^*$ will be mapped into a ellipsoid where the principal axes of the ellipsoid are given by the eigenvectors of the Gram matrix $J_{f^{-1}}(x^*)^T J_{f^{-1}}(x^*)$. Those eigenvectors are the singular directions of $J_{f^{-1}}(x^*)$ and correspond to direction of small and large varaibility if the corresponding singular value is small or large, respectively. For $J_{f^{-1}}(x^*)$ symmetric, the singular and eigenvalues coincide [3].

Thus, the flow of arguments from the simplest scenario takes on and we observe:

**Observation 1:** Large singular values of the NF's Jacobian evaluated at $x$ correspond to directions of small variability in the data. Directions of large variability, however, correspond to small singular values.

How can we make use of this observation to estimate the ID of a data manifold? When training an NF on clean manifold data (i.e. samples from a manifold without intrinsic noise), Observation 1 predicts a clear change in magnitude between singular values corresponding to off-manifold and on-manifold directions [4] . Indeed, we can observe such a change in magnitude in various toy examples, see Section 4.1. However, to estimate $d$ based on this observation only would require setting an arbitrary threshold on the magnitude of those singular values. On the other hand, in a more realistic scenario, real world data have some intrinsic noise.

**Data with intrinsic noise:** With the same setting as in the simplest scenario, we now additionally assume that the intrinsic noise is coming from a standard Gaussian with magnitude $\sigma_0$, i.e. $p(x) \sim \mathcal{N}(\mu, \Sigma + \sigma_0^2 I)$. Then, Equation (4) turns into

$$J_{f^{-1}}(x) = \text{diag}\left(\frac{1}{\sqrt{\sigma_0^2 + \sigma_1^2}}, \ldots, \frac{1}{\sqrt{\sigma_0^2 + \sigma_D^2}}\right). \tag{8}$$

Now, what happens if we inflate the data-manifold with Gaussian noise of variance $\sigma^2$? For directions of small variability in the data, $\sigma_i^2$ is small. Thus, only if $\sigma^2$ exceeds the intrinsic noise magnitude $\sigma_0^2$, we expect to see a change in $\lambda^{(i)}$. For directions of large variability, however, we expect to see a change in $\lambda^{(j)}$ only if $\sigma^2$ is greater than $\sigma_0^2 + \sigma_j^2$ as in this case $\sigma_j^2$ is not small. To account for the more general case where the intrinsic noise is not isotropic, we denote the intrinsic noise in the direction of the singular vector corresponding to singular value $\lambda^{(i)}$ as $\sigma_0^2(i)$ in the following.

**Observation 2:** If we were to inflate the noisy data-manifold with a Gaussian of variance $\sigma^2$ in direction of

(i) *small variability*, we expect the corresponding singular value $\lambda^{(i)}$ to change significantly only if $\sigma^2 \gg \sigma_0^2(i)$.

---

[3]Note, that the singular values are always positive and real, and therefore more convenient to analyze than the eigenvalues.

[4]Theoretically, eigenvalues/ singular values corresponding to off-manifold directions need to become infinity for $d-$dimensional manifolds. Indeed, training an NF on clean manifold data creates numerical instabilities as observed in [17, 20, 6].

(ii) *large variability*, we expect the corresponding singular value $\lambda^{(i)}$ to change significantly only if $\sigma^2 \gg \sigma_0^2(i) + \sigma_i^2$.

Indeed, we show in the supplementary that if the noise is only added in the manifold's normal space, then only those singular values will change while the singular values in direction of large variability will stay constant. Given this observation, we propose to train a separate NF for different noise magnitudes $\bar{\sigma}_1^2 < \cdots < \bar{\sigma}_N^2$, calculate for all of those models the flow's singular values on a sample $x$ from the manifold, and, according to Equation (8), find parameters $\alpha_i$ and $\beta_i$ such that $\hat{\lambda}(\sigma^2) = \frac{\beta_i}{\sqrt{\alpha_i{}^2 + \sigma^2}}$ fits the set of pairs $\{(\bar{\sigma}_n^2, \lambda_n^{(i)}(x))\}_{n=1,\ldots,N}$. Note that $\beta_i/\alpha_i$ corresponds to $\lambda^{(i)}$ if no noise was used to inflate the manifold and that $\alpha_i^2$ corresponds to the onset of strong decay.

**Our method for estimating the ID:** Given $(\alpha_i, \beta_i)$ for all $i = 1, \ldots, D$, we expect to see a clear pattern separating singular values corresponding to directions of small and large variability, respectively. According to Observation 1, $\beta_i/\alpha_i$ must be much larger for large variability directions and according to Observation 2, the onsets $\alpha_i^2$ must be smaller compared to small variability directions. More concretely, the latter suggests that counting the number of singular values with onset $\leq \alpha$,

$$F(\alpha) = \#\left\{i \in \{1, \ldots, D\} \text{ s.t. } \alpha^{(i)} \leq \alpha\right\}, \tag{9}$$

$F(\alpha)$ should have a plateau when varying $\alpha$ from $-\infty$ to $\infty$ corresponding to the number of singular values in direction of small variability. Ideally, this plateau can be seen clearly when plotting $F(\alpha)$ and thus the number of off-manifold directions can be directly read out.

Alternatively, a sum of two sigmoidal functions can be fitted to approximate $F$,

$$\hat{F}(\alpha) = \frac{a_1}{1 + e^{-b_1(\alpha - c_1)}} + \frac{a_2}{1 + e^{-b_2(\alpha - c_2)}}, \tag{10}$$

where $c_1 < c_2$ and hence $a_1$ corresponds to the number of small variability directions and $a_2$ to the number of large variability directions, see Algorithm 4 for an algorithmic overview of the method.

---

**Algorithm 4:** Estimating the intrinsic dimensionality given a set of data points.

---

**Require:** Dataset $\mathcal{D} = \{x_i\}_{i=1}^{M}$ with $x_i \in \mathbb{R}^D$. Samples $\{x_i^*\}_{i=1}^{K} \subset \mathcal{D}$ to use for estimating the ID. Noise magnitudes $\bar{\sigma}_1^2 < \ldots < \bar{\sigma}_N^2$.

    **for** $n = 1$ **to** $N$ **do**
        $\rightarrow$ add noise to dataset: $\tilde{\mathcal{D}} = \{x_i + \varepsilon_i\}$, where $\varepsilon_i \sim \mathcal{N}(0, \bar{\sigma}_n^2 I_D)$
        $\rightarrow$ learn $q_{\bar{\sigma}_n^2}$ using an NF $f_n$
        $\rightarrow$ calculate and store singular values $\{\lambda_n^{(i)}(x_k^*)\}_{i=1,\ldots,D}$ of $J_{f_n^{-1}}(x_k^*)$, $k = 1, \ldots, K$
    **end for**
    $\rightarrow$ calculate mean singular values for all $i$ and $n$, $\hat{\lambda}_n^{(i)} := \frac{1}{K} \sum_{k=1}^{K} \lambda_n^{(i)}(x_k^*)$
    **for** $i = 1$ **to** $D$ **do**
        $\rightarrow$ store $(\alpha_i, \beta_i)$ such that $\hat{\lambda}(\sigma^2) = \frac{\beta_i}{\sqrt{\alpha_i{}^2 + \sigma^2}}$ fits $\{(\bar{\sigma}_n^2, \hat{\lambda}_n^{(i)})\}_{n=1,\ldots,N}$
    **end for**
    $\rightarrow$ calculate $F(\alpha) = \#\{i \text{ s.t. } \alpha_i \leq \alpha\}$ for $\alpha \in (-\infty, \infty)$
    $\rightarrow$ fit $\hat{F}(\alpha)$ given by Equation (10) and store parameter $a_2$
    $\rightarrow$ estimator is given by $\hat{d} = a_2$

---

The averaging over the singular values does not have a geometrical interpretation as such. We are only interested in the cut-off separating singular values corresponding to on- and off-manifold directions. Hence, the averaging reduces the noise introduced by limited sample size or unexact learning of $q_{\sigma^2}$.

## 4.1 Pedagogical example

We illustrate our method on a pedagogical example - a uniform distribution on the unit sphere corrupted by some intrinsic Gaussian noise with magnitude $\sigma_0^2 = 10^{-6}$. In Figure 1 left, we show how the three singular values change in log-log-scale depending on various noise magnitudes. For the error bars, we use 200 clean training examples (i.e. not corrupted by noise). After the intrinsic noise magnitude is surpassed (first dashed line from the left), the large singular value changes with slope $\approx -0.5$ whereas the two smallest singular value barely change. After the noise magnitude

exceeds the radius of the sphere (second dashed line from the left), the singular values approach each other. In the right figure, we show $F(\alpha)$ and its estimate $\hat{F}(\alpha)$. The title of the right figure shows the output of Algorithm 4, the parameter $a_2$ of $\hat{F}(\alpha)$.

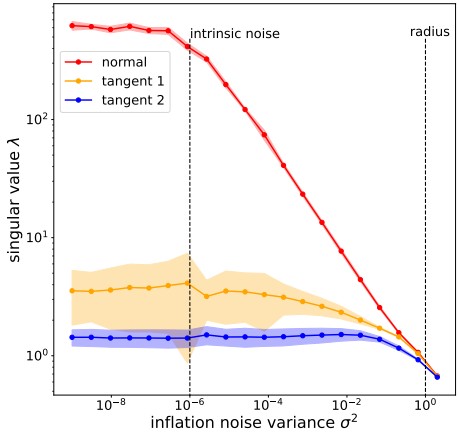 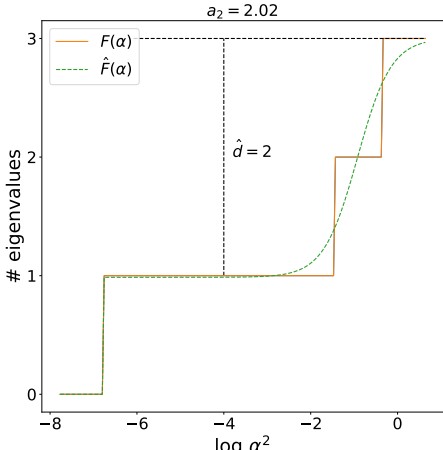

Figure 1: **Left:** $\lambda$ vs. $\sigma^2$ plot as described in the main text. The red curve corresponds to singular values associated to direction of small variability (i.e. normal to the manifold). The blue and yellow curves correspond to singular values associated to directions of large variability (i.e. tangent to the manifold). **Right:** $F(\alpha)$ and $\hat{F}(\alpha)$ as defined in Equations (9) and (10).

## 4.2 Special case: Images

So far, our theory assumes unbounded data such that if the inflation noise $\sigma^2$ tends to infinity, $\sigma^2 \to \infty$, the eigenvalues of the flow's Jacobian tend to $0$, $\lambda \to 0$. According to our theory, this decay happens much later for singular values corresponding to manifold directions than for off-manifold directions manifesting as a plateau for the decay onsets $\alpha$. According to Equation (8), those onsets reflect the amount of variability in the manifold directions.

However, some real world datasets are bounded. For example, RGB images consist of pixels which take values in $[0, 255]$. Thus, for $\sigma^2 \to \infty$, inflated images will become too noisy and all the information about the manifold is lost. Now, assume that $\sigma^2_{\max}$ is the maximal amount of noise that can be tolerated before all the images become so noisy that they become meaningless. Then, all onsets greater or equal to $\sigma^2_{\max}$ will correspond to manifold directions as the amount of variability in manifold directions must be greater than $\sigma^2_{\max}$.

How can we set this $\sigma^2_{\max}$ in the least arbitrary way? In the supplementary, Section A.4, we compute this maximal amount of noise from the following rationale. If $\sigma^2$ is too large, then essentially all the pixels will saturate to either the lower or the upper bounds, so a reasonable maximal amount of noise is such that 50% of the pixels are saturated. If the pixel values fall in the interval $[0, x_{\max}]$ (and if we assume that pixel values are uniformly distributed on this interval), then $\sigma_{\max} = 0.68 x_{\max}$.

## Remark 1

(i) *Note, for $K = 1$, Algorithm 4 estimates the ID locally at a given point $x_1^*$. In the supplementary, we show that we can learn the ID locally on the lolipop dataset proposed in [32]. This manifold consists of a 1 dimensional line (the stick of a lolipop) and a 2 dimensional disk.*

(ii) *There is no need to calculate $F(\alpha)$ for all $\alpha \in \mathbb{R}$ since $F(\alpha) \in \{0, 1, \ldots, D\}$. Given $\alpha_i$ for all $i = 1, \ldots, D$, we set $\alpha \in [\alpha_{min}, \alpha_{max}]$ where $\alpha_{min} = argmin_{i=1,\ldots,D}\{\alpha_i\}$ and $\alpha_{max} = argmax_{i=1,\ldots,D}\{\alpha_i\}$.*

| Distribution | D | ID | ID-NF | LIDL | twoNN |
|---|---|---|---|---|---|
| mixture on sphere | 3 | 2 | 2.01 | 1.91±0.06 | 1.98 |
| correlated on sphere | 3 | 2 | 2.04 | 1.66±0.07 | 1.99 |
| mixture on torus | 3 | 2 | 2.02 | 2.05±0.04 | 1.97 |
| correlated on torus | 3 | 2 | 2.02 | 2.07±0.05 | 2.02 |
| correlated on hyperboloid | 3 | 2 | 2.02 | 2.01±0.07 | 1.99 |
| unimodal on hyperboloid | 3 | 2 | 2.01 | 1.92±0.1 | 1.96 |
| exponential on thin spiral | 2 | 1 | 1 | 1.08±0.06 | 1 |
| mixture on swiss roll | 3 | 2 | 2.02 | 2.26±0.03 | 1.98 |
| correlated on swiss roll | 3 | 2 | 2.02 | 2.48±0.03 | 1.94 |
| mixture on stiefel | 4 | 1 | 1.07 | 1.19±0.01 | 0.99 |

Table 1: Performance of different ID estimation methods (ID-NF, LIDL, twoNN), on various datasets. Numbers highlighted in orange depict strong deviation from ground truth.

## 5 Experiments

We benchmark our method with twoNN and LIDL, see Section 3. The latter is very similar to our method as we both inflate the manifold with different values of $\sigma^2$ and then use an NF to learn $q_{\sigma^2}$. However, [32] estimate the rate at which the log-likelihoods change depending on the noise magnitude $\sigma^2$, whereas we study how the flow's Jacobian eigenvalues evolve. Hence, we rely on the NF being able to transform a sample of $p(x)$ to a standard Gaussian - a much easier task than learning the inflated distribution $q_{\sigma^2}$ exactly. Also, we don't rely on the assumption that $q_{\sigma^2}(x) = p(x)\mathcal{N}(x; x, I_{D-d})$ and thus we don't need to fine-tune the different noise magnitudes $\bar{\sigma}_1^2, \ldots, \bar{\sigma}_N^2$, see Section 3 for more details.

As an abbreviation for our method, we use ID-NF. We refer to the supplementary for corresponding training details and additional figures. The code for using ID-NF or reproducing our experiments can be found here `https://github.com/chrvt/ID-NF`.

### 5.1 Low-dimensional synthetic datasets

We test our method on various synthetic datasets with known ID: a sphere, torus, hyperboloid, thin spiral, swiss roll, and Stiefel manifold, see Table 1. We use different distributions on those manifolds to test the sensitivity on the sampling distribution $p(x)$. We compare our method with the LIDL and twoNN estimator introduced in Section 3. For those low-dimensional examples, all methods perform well. The twoNN and our method estimate all ID exactly. The LIDL method, however, shows some sensitivity towards the sampling distribution (see the highlighted estimate for the correlated on sphere distribution), and slightly overestimates the swiss roll dimensionality (see the highlighted estimate for the swiss roll). However, we did not try to find the optimal range for $\sigma^2$ and use the same across all distributions. By this, we want to demonstrate the disadvantage of having to estimate the correct range for every distribution separately.

### 5.2 High-dimensional synthetic datasets

Next, we study how the methods scale to higher embedding dimensions. For that, we sample uniformly from $\mathcal{S}(D/2)$ embedded in $\mathbb{R}^D$ for different even values of $D$. For a fair a comparison, we only use a training set of size $10^4$ for all methods. In Figure 2 left, we display how the different methods estimate the dimensionality. ID-NF is on par with LIDL and outperforms twoNN. The latter suffers from the curse of dimensionality as mentioned in the introduction, Section 1.

In Figure 2 right, we repeat the experiment using only $10^3$ samples. ID-NF still performs very well. LIDL, however, has greater variability and significantly underestimates the ID for $d = 200$ (i.e. $D = 400$). This indicates another benefit of our method: as opposed to LIDL, we do not need to learn the density exactly, a task that requires more samples for higher dimensions.

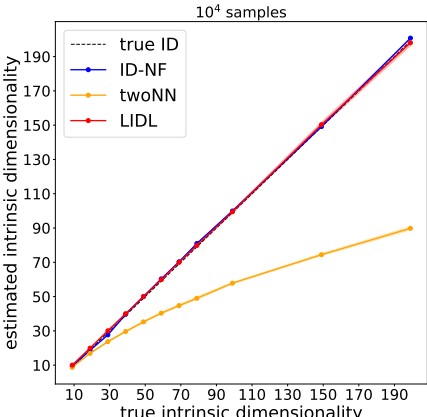
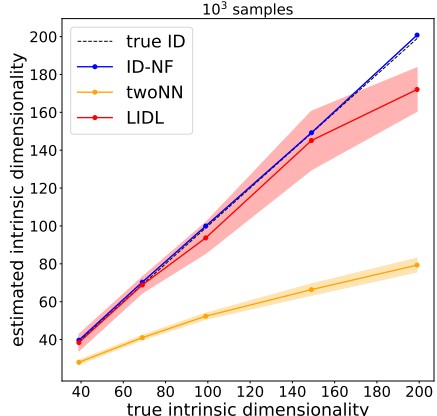

Figure 2: **Left:** Performance of different ID-estimation methods on uniform samples from $\mathcal{S}(D/2)$ embedded in $\mathbb{R}^D$ for $D$ ranging from 20 to 400. **Right:** Corresponding plot using $10^3$ instead of $10^4$ samples for traning.

### 5.3 StyleGan image manifolds

In [3], a image manifold was created by using a recent variant of a generative adversarial network, StyleGan2 [19], trained on a high-quality image dataset, the FFHQ dataset [18]. When generating new images, only $d$ of total $512$ latent variables were varied while keeping the others fixed. Note, that this does not necessarily generates a $d-$dimensional image manifold. Indeed, [29] showed that whenever the generater is Lipschitz continuous, the ID is at most $d$. We downsample those images to a resolution of $64 \times 64 \times 3$, i.e. the embedding space has dimension $D = 12288$.

We apply our method on the StyleGan $d = 2$ and $d = 64$ image manifold consisting of $10^4$ and $2 \cdot 10^4$ images, respectively. Unfortunely, training $N$ flows on these datasets is computationally expensive, so is calculating the eigenspectrum of the flows Jacobian for $K$ samples. However, we are only interested whether the onset of the eigenvalue's decay happens after $\sigma_{\max} = 255 \cdot 0.68$ or not, see Section 4.2. Therefore, it is sufficient to train only 3 NFs: one where the inflation noise $\sigma$ has a very small magnitude, one where $\sigma = 255 \cdot 0.68$, and one where $\sigma$ is very large.

In Figure 3, we show the heights and onsets for all singular value curves of a specific example for $d = 2$ on the left, and $d = 64$ on the right. We highlight the points corresponding to the 2 smallest and 64 smallest singular values, respectively, in red. The dashed vertical line is located at $\sigma_{\max} = 255 \cdot 0.68$. For $d = 2$, the smallest two singular values are nicely separated, though we count $\hat{d} = 4$ singular values which have onsets greater than $\sigma_{\max}^2$ for this particular example. For $d = 64$, we count $\hat{d} = 57$. When averaging over $K = 50$ samples, our estimate for $d = 2$ is $\hat{d} = 4.06$ and for $d = 64$ we have $\hat{d} = 62.24$.[5]

We also estimated the ID using LIDL, however, we did not get a consistent estimator. In fact, depending on the range one uses for the inflation noise $\sigma^2$, the estimate varies greatly.

### 5.4 Proof-of-concept applications

We show that the ID can be used to improve latent variable models and observe that out-of-distribution (OOD) samples have a higher ID.

**Latent variable models:** Recently, two latent variable models for manifold valued data based on NFs were developed, the manifold flow ($\mathcal{M}-$flow) [3] and denoising normalizing flow (DNF) [16]. Both methods rely on knowing the exact number of latent variables (i.e. the ID) which limits their applicability to real-world problems. For instance, the true ID (if exists) of the CelebA-HQ [18] is

---

[5]We do not average over the singular values since we are not interested in the cut-off separating eigenvalues corresponding to on- and off-manifold directions. We estimate $d$ for each sample individually, and then average.

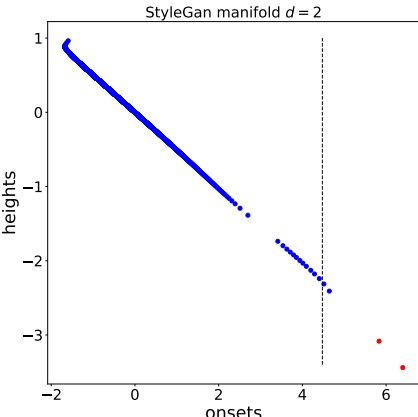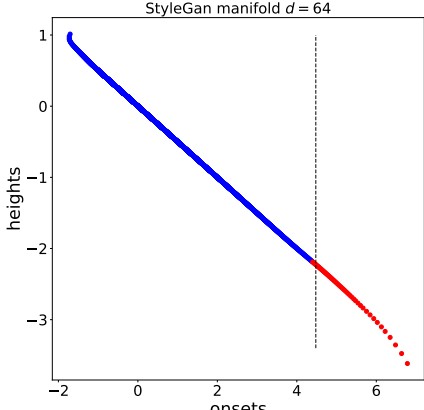

Figure 3: Scatter plots of onsets and heights, $(\alpha, \beta/\alpha)$, for all singular values in log-log scale. In red, the $d$ smallest singular values are highlighted. The dashed line corresponds to the $\sigma_{\max} = 0.68 \cdot 255$.

unknown and the latent dimensionality was set arbitrarily to 512. We used our method to estimate the ID and obtained $\hat{d} = 130$. We then trained an DNF with the same architecture as in the original paper using a latent dimensionality of 130 instead of 512. After 300 epochs, we assess the quality of the generated images with the Frechet Inception distance (FID) - where lower distances are better [15]. We obtain an FID score of 36.92. The original DNF has an FID score of 34.14, and the $\mathcal{M}-$flow of 38.07 - after 500 epochs of training. Thus, we obtain very similar results in terms of generative power (measured by the FID score) with only 130 latent dimensions instead of 512.

**ID on OOD-samples:** Having trained on a specific dataset, such as the StyleGan 2d image manifold, how does the ID changes for an out-of-distribution (OOD) sample? In table 2, we report the average ID estimated on $K = 50$ samples from different datasets when trained on the Stylegan2d image manifold. As we can see, the ID of OOD samples is significantly higher than for in-distribution samples. Intuitively, OOD samples live on the manifolds normal space, and thus, in case $d < D/2$, have more directions of large variability. We conduct more OOD experiments in the supplementary.

| Datasets / OOD | StyleGan2d | StyleGan64d | CelebA |
|---|---|---|---|
| StyleGan2d | $4.06 \pm 1.75$ | $151.72 \pm 54.04$ | $272.14 \pm 261.71$ |

Table 2: Average ID estimates for 50 samples using our method trained StyleGan2d.

## 6 Discussion

We have introduced a new method to estimate the ID exploiting the ability of NFs to transform data into samples from a Gaussian random variable. Based on some simple back-of-the-envelope calculations, we derived how the singular values of the flows Jacobian evolve when inflating the data with Gaussian noise before training. Crucially for our estimator, singular values corresponding to the directions of large variability (i.e. manifold directions) evolve significantly different compared to singular values corresponding to the directions of small variability (i.e. off-manifold directions).

We demonstrated that we can estimate the ID for different manifolds with different sampling distribution. We compared our method to a state-of-the-art ID estimator based on nearest neighbor statistics, twoNN, and to a related method which is also based on NFs, LIDL. We outperform twoNN for high dimensions (Table 1), and LIDL for small data regimes (Figure 2) and images, Section 5.3. As opposed to LIDL, we don't have to fine-tune the noise magnitudes $\bar{\sigma}_1^2, \ldots, \bar{\sigma}_N^2$. However, we need $\bar{\sigma}_1$ to be small and $\bar{\sigma}_N$ to be sufficiently large which may be computationally expensive.

We showed that our method scales to RGB images of resolution $64 \times 64$ populating a $d = 2$ and $d = 64$ dimensional manifold, where we estimate $\hat{d} = 4$ and for $\hat{d} = 62$, respectively. However,

this estimate is sensitive to the number $K$ of images used to estimate $d$ (we used $K = 50$), the number of NFs trained on (we argued that $N = 3$ is sufficient), and to the maximum amount of noise to be tolerated before the images become too blurry ($\sigma_{\max} = 0.68 \cdot 255$ see Secetion 4.2). Neverthelesse, to the best of our knowledge, this is the first method which estimates the ID on such high-resolution image manifolds consistently. Additionally, we demonstrated that estimating the ID can help to improve recently developed latent variable models based on NFs where knowing this exact number is crucial. Also, we observed that for out-of-distribution (OOD) data, the ID is higher than for on-manifold examples motivating further research on the relation between OOD samples and ID.

Finally, our theoretical derivations rely on the assumption that data is unbounded, although we adapted our methiod for bounded data too. However, for this case more research is needed to understand the exact behaviour of the flow's Jacobian singular values when approaching the boundary. An interesting direction to pursue is to adapt the target distribution and inflation noise to the data topology. For images, this amounts to changing the NF's target distribution to be uniform instead of Gaussian, and using uniform instead of Gaussian noise.

**Broader Impact:** As a theoretical paper, we don't foresee any direct negative social impacts of our work.

## Acknowledgments and Disclosure of Funding

This study has been supported by the Swiss National Science Foundation grant 31003A_175644.

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
