# Estimating the intrinsic dimensionality using Normalizing Flows - Supplementary

## A  Theoretical appendix

### A.1  Singular value evolution under normal space noise

In [4], it was shown that if noise is only added in the manifold's normal space, then we have that $q_{\sigma^2}(x) = q(x|x)p(x)$ where $q(x|x)$ is the normalization constant of a $(D - d)-$dimensional Gaussian random variable with co-variance matrix $\sigma^2 I_D$. However, $\sigma^2$ needs to be sufficiently small, a condition formalized as $Q-$normal reachability. With these conditions, a direct consequence is that the singular values in on-manifold directions will not depend on $\sigma^2$. We adapt notations from the proof of the main Theorem in [4].

**Proposition 1** *Let $\mathcal{M}$ be a $d-$dimensional, $C^2$ manifold. For each $x \in \mathcal{M}$, let $q_{\sigma^2}(\cdot|x)$ denote a noise distribution with support in the normal space $N_x$ of $x$. Assume that there exists $\sigma_{\min}^2$ and $\sigma_{\max}^2$ with $0 \le \sigma_{\min}^2 < \sigma_{\max}^2$ such that we can learn the inflated distribution $q_{\sigma^2}(\tilde{x})$ exactly for all $\sigma^2 \in [\sigma_{\min}^2, \sigma_{\max}^2]$ using an NF $\tilde{f}_{\sigma^2}$ with standard Gaussian as reference distribution. Denote by $\lambda^{(i)}(\sigma^2)$ the $i-$th singular value of the Jacobian of $f_{\sigma^2}$ evaluated at a given point $x \in \mathcal{M}$. Then for all $x \in \mathcal{M}$ for all $\sigma^2 \in (\sigma_{\min}^2, \sigma_{\max}^2)$ it holds that $\frac{d\lambda^{(i)}}{d\sigma^2} = 0$ if and only if $\lambda^{(i)}$ is a singular value associated to on-manifold direction.*

**Proof of Proposition 1**

Let $x \in \mathcal{M}$ and $\sigma^2 \in (\sigma_{\min}^2, \sigma_{\max}^2)$. Since $\mathcal{M}$ is a $d-$dimensional $C^2$ manifold, there exists an open neighborhood $\mathcal{B}_x$ of $x$ in $\mathcal{M}$, an open set $\mathcal{U}_x$ in $\mathbb{R}^d$, and an invertible map $f : \mathcal{U}_x \mapsto \mathcal{B}_x, \mathcal{U}_x \subset \mathbb{R}^d$, such that $f$ and $f^{-1}$ are twice continuously differentiable. It follows that the Gram determinant of $J_f$ is non-zero for all $x \in \mathcal{B}_x$, i.e. $\det J_f(f^{-1}(x))^T J_f(f^{-1}(x)) \ne 0 \; \forall x \in \mathcal{B}_x$. We exploit this by constructing a local diffeomorphism $\tilde{f}$ in the following.

For that we denote by $A_u$ the matrix with columns consisting of normal vectors spanning the normal space in $x = f(u), u \in \mathcal{U}_x$. With $\mathcal{V}_x \subset \mathbb{R}^{D-d}$, we define $\tilde{f} : \mathcal{U}_x \times \mathcal{V}_x \subset \mathbb{R}^d \times \mathbb{R}^{D-d} \to \widetilde{\mathcal{B}}_x$ for some $\widetilde{\mathcal{B}}_x \subset \widetilde{\mathcal{X}}$ as follows:

$$\tilde{f}(u, v) = f(u) + \sigma A_u v. \tag{1}$$

Since the Gram determinant of $f(u)$ is non-zero, $\tilde{f}$ is a diffeomorphism which follows from the inverse function theorem, see proof of the main Theorem in [4]. Note that $q_{\sigma^2}$ is uniquely determined by the latent distribution and the embedding $\tilde{f}$. Hence, if we fix the latent distribution to be standard Gaussian, we have that the NF used to learn $q_{\sigma^2}$ must be $\tilde{f}$ for all $(u, v)$, i.e. $\tilde{f}_{\sigma^2}(u, v) = \tilde{f}(u, v), \forall(u, v) \in \mathcal{U}_x \times \mathcal{V}_x$.

The Jacobian of $\tilde{f}$ is given by

$$J_{\tilde{f}}(u, v) = \left[ \; J_f(u) + \sigma \frac{\partial}{\partial u} A_u v \; \vdots \; \sigma \, A_u \; \right] \tag{2}$$

where $\frac{\partial}{\partial u}$ denotes the Jacobian of a function depending on $u$, and the dashed line seperates two block matrices. Evaluated at $(u, 0)$, we have that

$$J_{\tilde{f}}(u,0) = \left[\ J_f(u)\ \vdots\ \sigma A_u\ \right] \tag{3}$$

with the columns of $J_f(u)$ spanning the tangent space at $x$ and the columns of $A_u$ the normal space. The singular values of $J_{\tilde{f}}(u,0)$ are given by the eigenvalues of

$$J_{\tilde{f}}(u,0)^T J_{\tilde{f}}(u,0) = \left( \begin{array}{c|c} J_f(u)^T J_f(u) & \sigma J_f(u)^T \cdot A_u \\ \hline \sigma A_u^T \cdot J_f(u)^T & \sigma^2 A_u^T A_u \end{array} \right)$$

$$= \left( \begin{array}{c|c} J_f(u)^T J_f(u) & 0_{d \times D-d} \\ \hline 0_{D-d \times d} & \sigma^2 A_u^T A_u \end{array} \right) \tag{4}$$

$$\tag{5}$$

where in the last step we have exploited that the columns of $J_f$ and $A_u$ are orthogonal. Therefore, the eigenvalues of $J_{\tilde{f}}(u,0)^T J_{\tilde{f}}(u,0)$ consists of the $d$ eigenvalues of $J_f(u)^T J_f(u)$ and the $(D-d)$ eigenvalues of $\sigma^2 A_u^T A_u$. Therefore, we have that $\frac{d\lambda^{(i)}}{d\sigma^2} = 0$ if and only if $\lambda^{(i)}$ is an eigenvalue of $J_f(u)^T J_f(u)$. However, these eigenvalues are exactly in direction of large variability, i.e. in on-manifold direction. This was to be shown.

## A.2 Complexity of Algorithm 1

The complexity of the algorithm depends on a) the complexity to train $N$ NFs, b) calculating the Jacobian of $K$ samples, and c) calculating the singular value decomposition of the Jacobian.

The complexity of a) depends on the choosen architecture for the NFs. It is important to note that for bounded data, such as images, $N = 3$.

The complexity of b) depends on the operational complexity of the NF i.e. the number of operations needed for one forward pass, see [10].

The complexity of c) is given by $\mathcal{O}(D^3)$.

## A.3 Convergence of Algorithm 1

Let $p(x) = \mathcal{N}(0,\Sigma)$ be a Gaussian in $\mathbb{R}^d$ and denote the ordered eigenvalues of $\Sigma$ as $\sigma_1^2 \leq \ldots \leq \sigma_d^2$. Now, we embedd $\mathbb{R}^d$ into $\mathbb{R}^D$ by padding the missing $(D-d)$ coordinates with zeros, and add isotropic Gaussian noise with co-variance $\sigma^2 I_D$. Then, as we have seen in Section 4 of the main text, the true NF $f$ transforming $\tilde{x} \sim q(\tilde{x})$ into $u = f^{-1}(\tilde{x})$ distributed according to $\mathcal{N}(0,I_D)$ is given by

$$f^{-1}(x) = \bar{\Sigma}^{-\frac{1}{2}}(x - \mu) \tag{6}$$

with Jacobian

$$J_{f^{-1}}(x) = \bar{\Sigma}^{-\frac{1}{2}} = S^T D^{-\frac{1}{2}} S \tag{7}$$

where $D^{\frac{1}{2}} = \mathrm{diag}(\sqrt{\sigma_1^2 + \sigma^2}, \ldots, \sqrt{\sigma_d^2 + \sigma^2}, \sigma, \ldots, \sigma)$ and $S$ is orthonormal and consists of the eigenvectors of $\bar{\Sigma}$. Thus, the eigenvalues of $J_{f^{-1}}(x)$ are given by

$$\lambda^{(i)}(\sigma^2) = \begin{cases} \frac{1}{\sqrt{\sigma_i^2 + \sigma^2}}, & \text{for } i = 1, \ldots, d \\ \frac{1}{\sigma}, & \text{else.} \end{cases} \tag{8}$$

Note that the function to learn these trajectories, $\hat{\lambda} = \frac{\beta}{\sqrt{\alpha^2 + \sigma^2}}$, is convex in $\alpha$. Therefore, we assume without loss of generality that $\alpha_i = \sigma_i$ for all $i = 1, \ldots, d$, and $\alpha_i = 0$ for all $i = d+1, \ldots, D$. Remember that

$$F(\alpha) = \#\left\{ i \in \{1, \ldots, D\} \text{ s.t. } \alpha^{(i)} \leq \alpha \right\}, \tag{9}$$

thus $F(\alpha)$ counts the number of eigenvalues with decay onset $\leq \alpha$, and we want to find the manifesting plateau.

Let us assume that $\sigma_1^2 = \cdots = \sigma_d^2$ in the following. Then, $F(\alpha)$ is the sum of two step functions where one step is at $\alpha = 0$ with height $(D-d)$, and the other step is at $\alpha = \sigma_1$ with height $d$. Note that a single step function can be learned arbitrarily well with a sigmoid function $\mathrm{sigmoid}_1(\sigma^2)$,

$$\text{sigmoid}_1(\alpha) = \frac{a_1}{1 + e^{-b_1(\alpha - c_1)}}, \tag{10}$$

if $b_1 \to \infty$, see [7]. Thus, in the case where $\sigma_1^2 = \cdots = \sigma_d^2$, we can learn $F(\alpha)$ arbitrarily well with the sum of two sigmoid functions $\text{sigmoid}_1(\alpha) + \text{sigmoid}_2(\alpha)$. Since the optimization problem can be easily turned into an optimazation problem which is convex in $a_1$ and $a_2$, we have that $a_1 = (D - d)$ and $a_2 = d$.

Now if $\Delta_i := \sigma_{i+1}^2 - \sigma_i^2 \geq 0$ is smaller than the length of the plateau, $\Delta_i \ll \sigma_1$, $a_1$ must still be $D - d$ and thus $a_2 = d$ as needed. Even if have some variation in the off-manifold directions, the same holds true as long as the differences are much smaller that the length of the plateau.

### A.4 Computing $\sigma_{\max}$

Our proposed method to estimate the intrinsic dimensionality for bounded data critically depends on the threshold $\sigma_{\max}$. Indeed all the eigenvalues having an onset $\alpha$ larger than $\sigma_{\max}$ are declared as eigenvalues associated to eigenvectors corresponding to directions of large variability (i.e. manifold directions). In this section, we compute this threshold in the presence of lower and upper bounds for the variable $x$. Loosely speaking, we want to compute the maximal amount of noise we can add to a bounded variable before it has a chance of 50% of hitting the bounds.

Practically, let us assume that the variable $x$ is bounded, i.e. $x \in [0, 1]$ (e.g. a pixel has a minimal and a maximal intensity). Adding a zero-mean Gaussian noise with standard deviation $\sigma$ to $x$ is not possible as such without some rectification. The noisy version $\tilde{x}$ of a pixel with value $x$ is given by

$$\tilde{x} = \max\left(\min\left(x + \epsilon, 1\right), 0\right) \quad \epsilon \sim \mathcal{N}(0, \sigma^2) \tag{11}$$

For a given $x$, the probability that $\tilde{x}$ hits one of the bounds is given by

$$
\begin{aligned}
p(\tilde{x} = 0 \text{ or } \tilde{x} = 1 | x) &= 1 - \int_0^1 \mathcal{N}\left(\tilde{x}; x, \sigma^2\right) d\tilde{x} \\
&= 1 - \frac{1}{2}\left(\text{erf}\left(\frac{1 - x}{\sqrt{2}\sigma}\right) - \text{erf}\left(-\frac{x}{\sqrt{2}\sigma}\right)\right) x
\end{aligned} \tag{12}
$$

where $\text{erf}(z) = \frac{2}{\sqrt{\pi}} \int_0^z \exp(-t^2) dt$ is the error function. If $p(x)$ denotes the distribution of pixel values, then the probability that a noisy pixel hits a bound is given by

$$f(\sigma) = \int_0^1 p(\tilde{x} = 0 \text{ or } \tilde{x} = 1 | x) p(x) dx \tag{13}$$

If we assume that $p(x) = \text{Uniform}([0, 1])$, we get

$$f(\sigma) = 1 - \text{erf}\left(\frac{1}{\sqrt{2}\sigma}\right) - \sqrt{\frac{2}{\pi}}\left(\exp\left(-\frac{1}{2\sigma^2}\right) - 1\right)\sigma \tag{14}$$

$\sigma_{\max}$ is defined as the value of $\sigma$ such that 50% of noisy pixels hit the bound, i.e.

$$\sigma_{\max} = f^{-1}\left(\frac{1}{2}\right) \simeq 0.68 \tag{15}$$

## B  Additional Experiments

### B.1  Lolipop

In [11], a manifold consisting of regions of different ID was considered - a 1 dimensional line segment, and a two dimensional disk such that the overall manfiold resembles a lolipop. The aim was to show that the LIDL method can esitmate the ID locally, although it was trained globally on all samples. We repeat this experiment and report the result in Figure S2. There, we display the estimated ID of 30 randomly selected examples together with 1000 samples from the dataset. The local ID is estimated correctly for both regions. However, note that by construction our estimate will never estimate $d = D$. Therefore, we embed the lolipop in $\mathbb{R}^3$ before training.

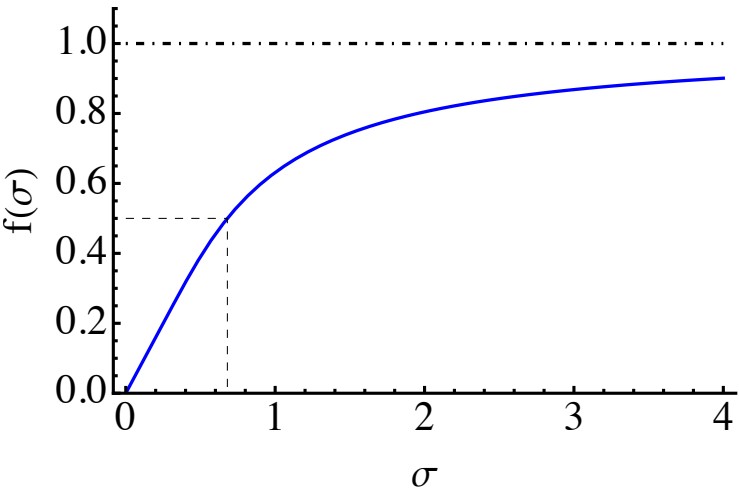

Figure S1: Fraction of pixels hitting the bound as a function of the noise level $\sigma$. $\sigma_{\max}$ is chosen such that $f(\sigma) = 0.5$.

## B.2 OOD detection

**OOD detection:** Having trained on a sepcific dataset, such as the StyleGan 2d image manifold, how does the ID changes for an out-of-distribution (OOD) sample? In table S1, we report the average ID estimated on $K = 50$ samples from different datasets. The rows show the datasets we trained on, while the columns represent the datasets on which we estimate the ID. As we can see, for the StyleGan2d and Stylegan64d the ID of OOD samples is significantly different than for in-distribution samples (note that StyleGan2d is included in StyleGan 64d). Thus, the ID could be used as a OOD detection method where samples with an ID different than the estimated one for in-distribution samples are classified as OOD. However, for the CelebA dataset as in-distribution, the difference is not significant anymore (though on average OOD samples have a higher ID). This is not suprising as it was already obseved in [6] that certain flow's architecture lead to learning local pixel correlations rather than semantic structure. Therefore, NFs trained on complex data will yield a similar likelihood value when evaluating on less complex data.

| Datasets / OOD | StyleGan2d | StyleGan64d | CelebA |
|:---:|:---:|:---:|:---:|
| StyleGan2d | $4.06 \pm 1.75$ | $151.72 \pm 54.04$ | $272.14 \pm 261.71$ |
| StyleGan64d | $-$ | $62.24 \pm 18.64$ | $137.04 \pm 37.1$ |
| CelebA | $144.08 \pm 15.22$ | $160.66 \pm 30.97$ | $126.62 \pm 26.5$ |

Table S1: Average ID estimates for 50 samples using our method trained and evaluated on different datasets (rows and columns, respectively).

## C  Training details

### C.1  Low-dimensional syntehtic datasets

We evaluate our method on different manifolds and distributions as proposed in [4], see Figure S3 ,S4 and S5 for a depiction of the different distributions (left column), evolution of singular values (middle column) and our estimate based on $F(\alpha)$ (right column). We refer to the Appendix B in [4] for technical details of the distributions. All the intrinsic dimensionalities are correctly retrieved. We repeated the experiments using uniform instead of Gaussian noise but did not find any significant difference. Though, when the flow is not expressive enough, we failed to estimate the true ID.

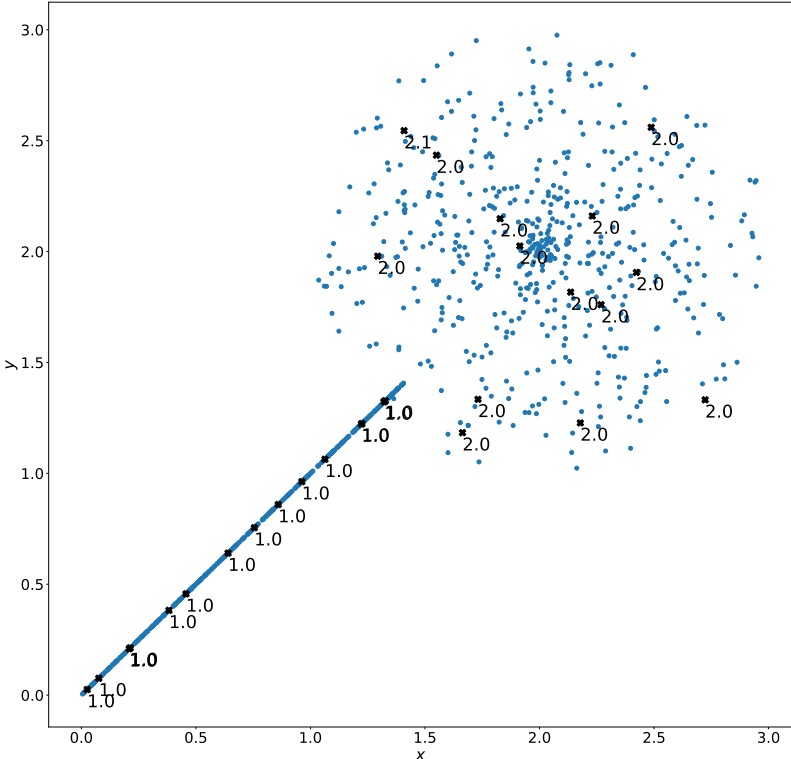

Figure S2: Samples from the lolipop dataset. **In black:** Test points on which the ID was estimated together with the value of the estimator.

As in [4], we use a Block Neural Autoregressive Flow (BNAF) [2] with 3 hidden layers consisting of 210 hidden dimensions each to train $q_{\sigma^2}$ for $\sigma^2 \in [\sigma_1^2, \sigma_N^2]$ using Adam optimizer [5]. We train for $N = 26$ different $\sigma^2$, with $\sigma_1^2 = 10^{-9}$ and $\sigma_N^2 = 10$.

## C.2 High-dimensional synthetic datasets

We use a BNAF, see Table C.2, to train $q_{\sigma^2}$ for $\sigma^2 \in [\sigma_1^2, \sigma_N^2]$ using Adam optimizer. We train for $N = 20$ different $\sigma^2$, with $\sigma_1^2 = 10^{-9}$ and $\sigma_N^2 = 2.0$. In Figure S6, we show the evolution of singular values together with our estimate based on $F(\alpha)$ as described in the main text for $D = 400$. The estimated dimensionality $\hat{d} = 200.7$ is very close to the ground truth ID ($d = 199$).

| Data dimension | hidden layers | hidden dimension | total parameters | epochs |
|---|---|---|---|---|
| 20 | 3 | 200 | 210,440 | 500 |
| 40 | 5 | 200 | 218,480 | 500 |
| 60 | 5 | 240 | 319,800 | 500 |
| 80 | 5 | 320 | 567,200 | 500 |
| 100 | 5 | 300 | 513,800 | 500 |
| 120 | 5 | 280 | 474,040 | 500 |
| 140 | 5 | 300 | 474,040 | 500 |
| 160 | 5 | 320 | 618,560 | 500 |
| 180 | 5 | 360 | 782,280 | 500 |
| 200 | 5 | 400 | 965,200 | 500 |
| 300 | 4 | 600 | 1,806,600 | 200 |
| 400 | 4 | 800 | 3,208,800 | 200 |

Table S2: BNAF details for circle experiments.

### C.3 StyleGan image manifolds

We use rational-quadratic splines [3] to train $q_{\sigma^2}$ for $\sigma \in [\sigma_1, \sigma_N]$ using AdamW optimizer [9] and cosine annealing [8]. We use the same settings as in [1] which also trained an NF on the StyleGan 2d and 64d manifolds. We train for $N = 3$ different $\sigma^2$, with $\sigma_1 = 10^{-9}$, $\sigma_1 = 255 \cdot 0.68$ and $\sigma_N = 255 \cdot 10^2$.

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

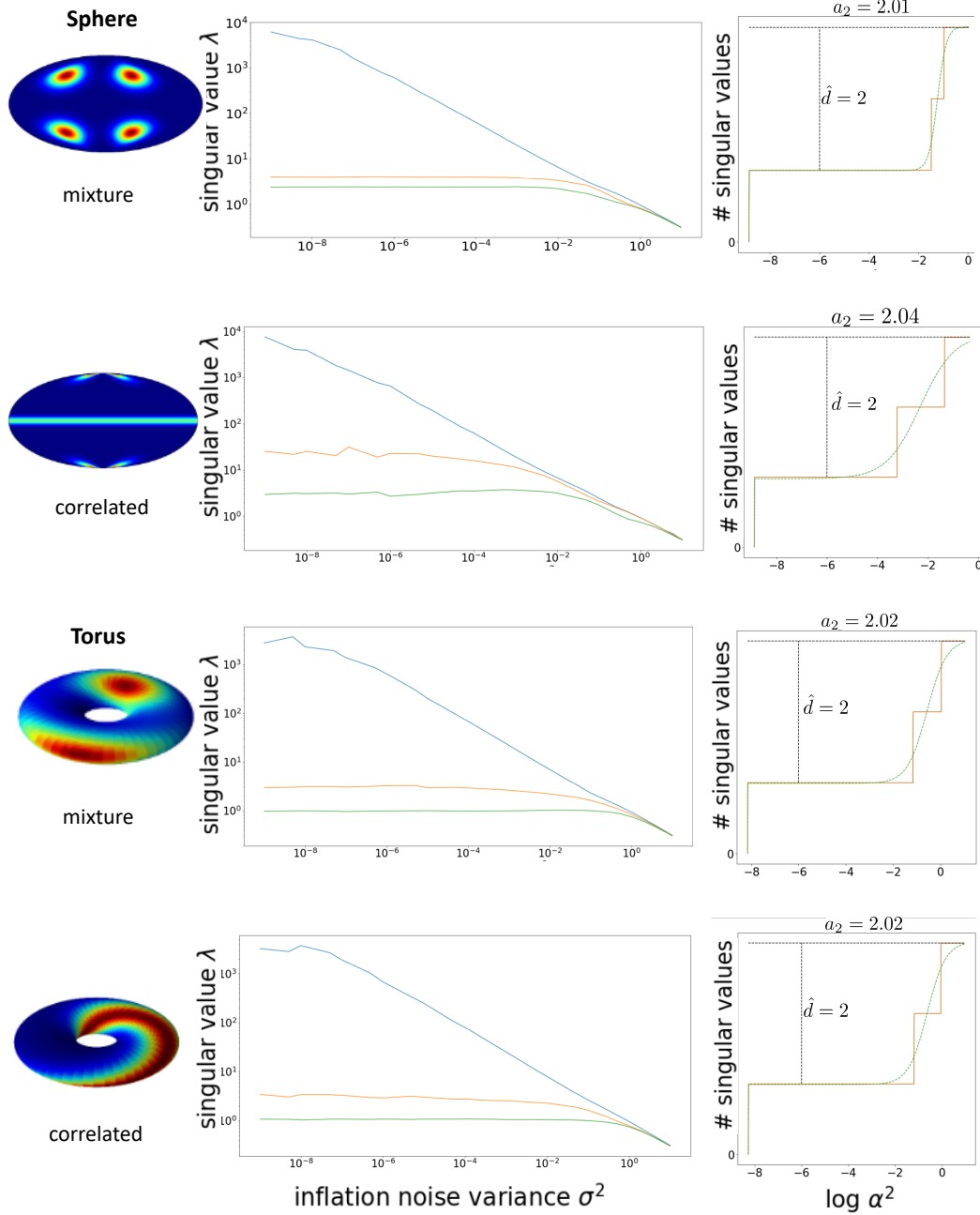

Figure S3: Mixture and correlated on sphere and torus (left column). Singular values as a function of $\sigma^2$ in log-log scale (middle column). $F(\alpha)$ in orange and $\hat{F}(\alpha)$ in dashed green (right column). In all the 4 examples the intrinsic dimension is correctly retrieved (i.e. $\hat{d} = 2$).

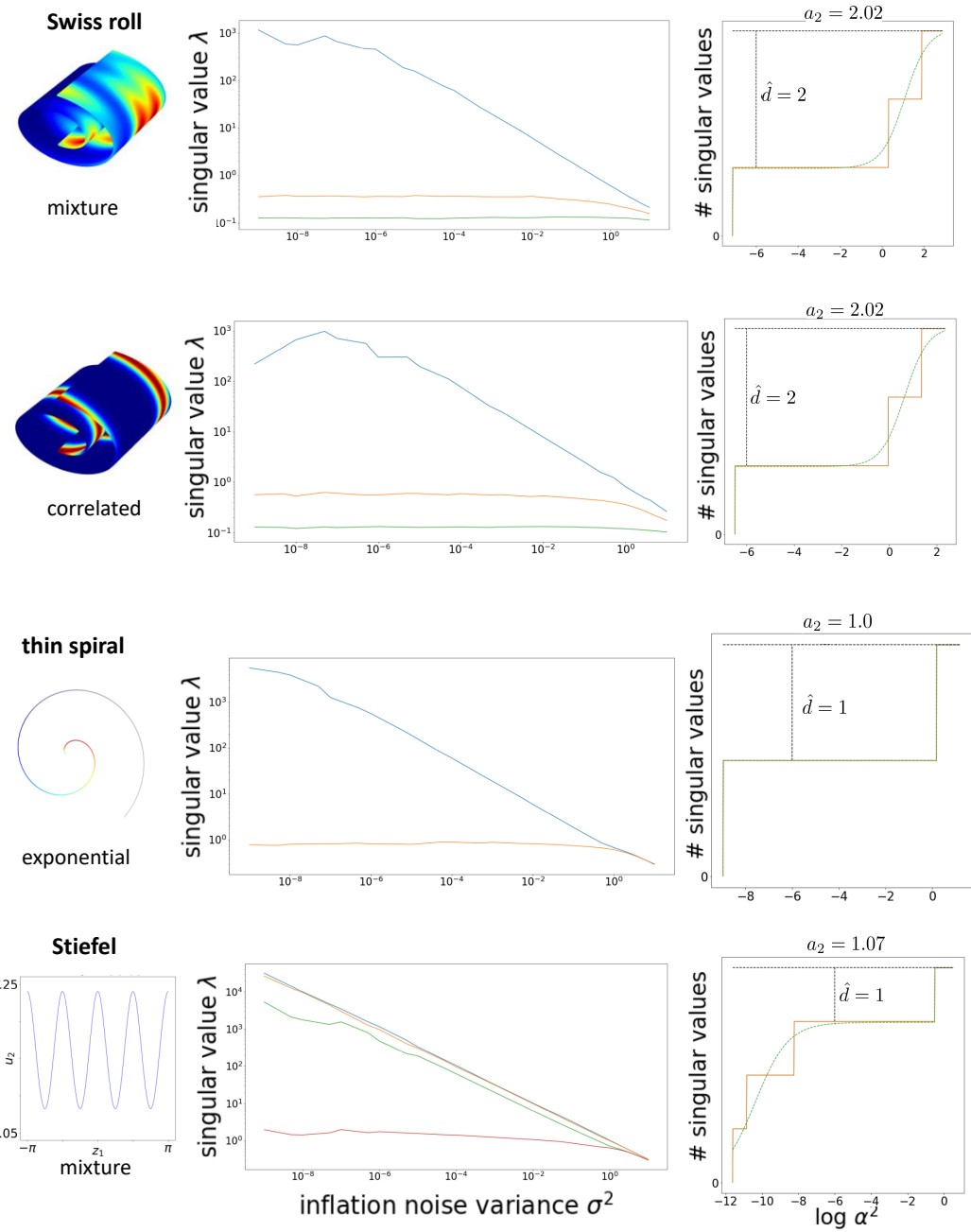

Figure S4: Mixture/unimodal and correlated on swiss roll/ hyperboloid (left column). Singular values as a function of $\sigma^2$ in log-log scale (middle column). $F(\alpha)$ in orange and $\hat{F}(\alpha)$ in dashed green (right column). Here also, in all the 4 examples the intrinsic dimension is correctly retrieved (i.e. $\hat{d} = 2$).

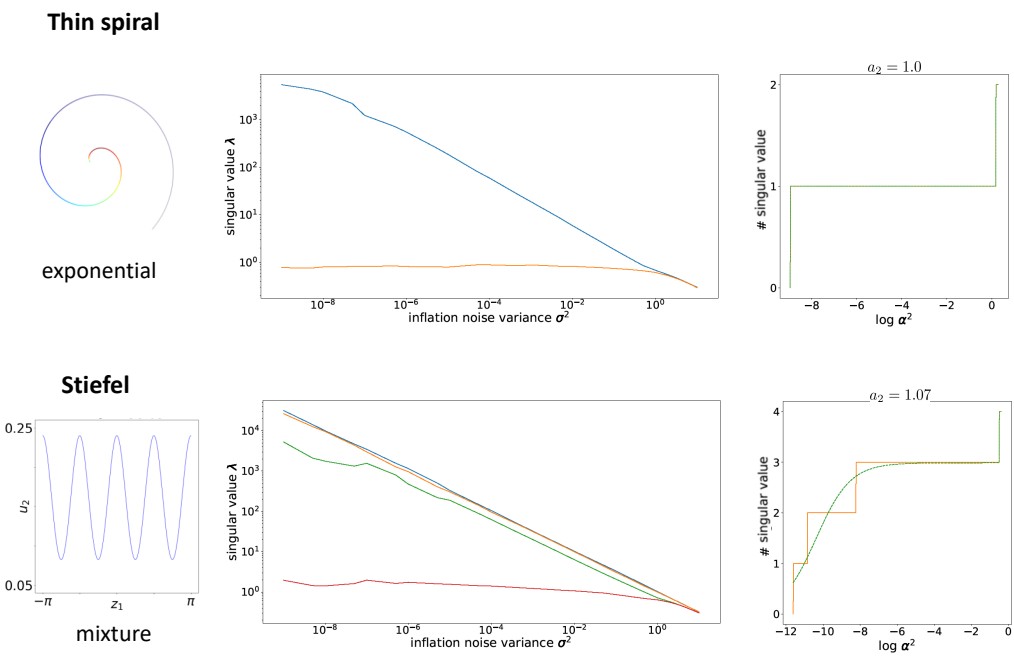

Figure S5: Exponential on thin spiral and the latent distribtuion used to generate points on the stiefel manifold (left column). Singular values as a function of $\sigma^2$ in log-log scale (middle column). $F(\alpha)$ in orange and $\hat{F}(\alpha)$ in dashed green (right column). The intrinsic dimension is correctly retrieved (i.e. $\hat{d} = 1$) for those two examples

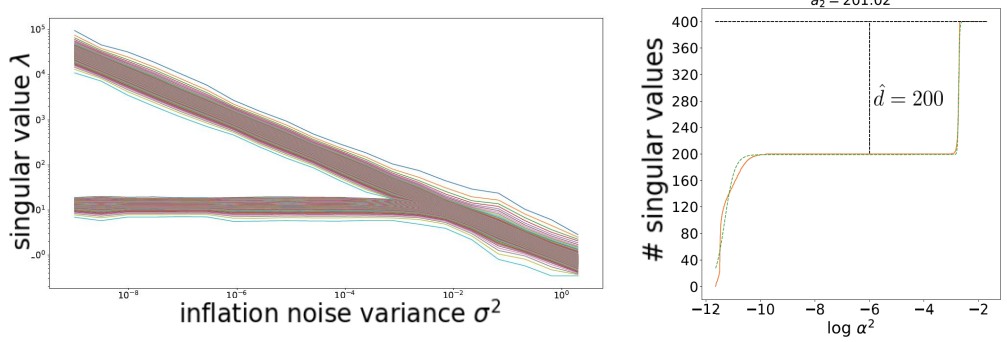

Figure S6: **Left:** Singular values as a function of $\sigma^2$ in log-log scale (middle column). **Right:** $F(\alpha)$ in orange and $\hat{F}(\alpha)$ in dashed green (right column).