# OpenReview forum: "Intrinsic dimensionality estimation using Normalizing Flows"
_NeurIPS.cc/2022/Conference — NeurIPS 2022 Accept_

### Official Review · Reviewer_aewG · 2022-07-10

**Rating:** 5
**Confidence:** 3
**Soundness:** 3 good
**Presentation:** 2 fair
**Contribution:** 3 good

**Summary:**

This paper proposes a new approach termed ID-NF to estimating intrinsic dimensionality based on normalizing flows which can scale to large datasets and high dimensions ($64\times64\times 3$).
ID-NF first specifies a dataset $\mathcal{D}=x_i, x_i\in \mathbb{R}^D$, $K$ samples $x_1^*, \dots, x_K^* \in \mathcal{D}$, and $N$ noise levels $\bar \sigma_1^2 <\dots <\bar\sigma_N^2$.
For each noise level, ID-NF injects noise $\epsilon_n\sim \mathcal{N}(0, \bar\sigma_n^2 I)$ to the data and use an NF $f_n$ to learn the distribution $q_{\bar\sigma_n^2}$. Then ID_NF calculate the eigenvalues ${\lambda_n^{(i)}(x_k^*)}$ of $J_{f^{-1}_n}(x_k^*)$ for each $k=1,\dots,K$ and compute $\hat\lambda_n^{(i)} = \frac1K \sum_1^K \lambda_n^{(i)}(x_k^*)$.
By fitting $\hat\lambda_n (\sigma^2) = \frac{\beta_i}{\sqrt{\alpha^2_i + \sigma^2}}$ with $\alpha_i, \beta_i$, ID-NF use {$\alpha_i$} to estimate intrinsic dimension.

**Questions:**

* I think Table1 is not a fair comparison, since the dimensionality estimated by ID-NF is naturally around an integer, but LIDL seems not.

**Limitations:**

Limitations seen above. Societal impact: none

**Strengths And Weaknesses:**

**Strengths**:
* To the best of my knowledge, the intrinsic dimensionality estimation method based on NFs is novel.
* Detailed experiments on pedagogical examples, low-dimensional synthetic examples, and high dimensional images generated by StyleGAN show the effects of this method.

**Weakness**:
* The paper approximate the NFs with $f^{-1}(x) \approx f^{-1}(x^*) + J_{f^{-1}}(x^*)(x-x^*)$, thus the distribution of $x$ should be approximately Gaussian. Then the Observation1 and 2 holds. This is valid locally, but not globally. I think this gap is not well addressed in the paper.
* Training NFs is expensive.  For $N$ noise levels, ID-NF should train $N$ NFs to approximate $q_{\sigma_n}$, which is quite expensive. Computing eigenvalues of $J_{f^{-1}}$ for many high-dimensional samples ($10^4$ for 2d-styleGAN  and $2\times 10^4$ for 64d-styleGAN) is also expensive.
* Algorithm4 computes mean eigenvalues of $J_{f^{-1}}$. I think for different samples $x_k^*$, eigenvalues of $J_{f^{-1}}(x^*_k)$ correspond to the variability along the directions indicated by eigenvectors, so how is the averaging operation performed?

**Minor**:
* Line188: grea*ter.
* Line159: As I understand, $\beta_i/\alpha_i$ should be larger for *small* variability directions?

---

> ### Author Response · Authors · 2022-07-31
> **Answer to aewG**
>
> Answer to Question
> 1. The ID-NF is not necessarily around an integer when using the estimated parameter $a_{2}$. The fact that the ID-NF is so close to an integer simply shows the robustness of the method (similar to twoNN which is not naturally around an integer either).  *Edit: However, for the special case of $D=2$ or $D=3$ where there are only 2 or 3 eigen/singular values it is indeed  close to an integer. We will include a footnote to clarify this*.
>
> Answer to Weakness comments:
>
> 1. Our experiments on various datasets with different distributions indicate that Observations 1 and 2 also hold if $p(x)$ is not approximately Gaussian. We show in our "Theoretical additions" answer, that samples from a flow are locally Gaussian distributed and therefore $p(x)$ does not need to be Gaussian globally for the applications of our Observations.
> 2. Indeed, training $N$ NFs is expensive - though for low-dimensional data it is very fast. As an orientation, training an NF on any non-image datasets takes only a few minutes on a GPU. The entire experiment on the sphere $S(200)$ embedded in $D=400$ experiment took only 10 minutes. For bounded data such as images, $N$ can be set to 3 as explained in the general answer.
> 3. We are not quite sure to fully understand the question/concern. Indeed, we average over the eigenvalues of different samples. Thus, we calculate the eigenvalues for $K$ samples, and average over all the $i-th$ eigenvalues to obtain $\hat{\lambda}^{(i)}$. We found that this leads to less variability in the $\lambda$ vs. $\sigma^{2}$ plot used to estimate the ID. Alternatively, one could also estimate $\hat{d}$ for each sample and then estimate $d$ by averaging over such estimates (as explained in Footnote 2). We found that the latter leads to greater variability in the estimate (though it still leads to a correct ID).

---

> > ### Comment · Reviewer_aewG · 2022-08-08
> > **Update**
> >
> > The question and the first weakness comment are well explained for me.
> >
> > For the last question, I mean that for different samples, when averaging the largest eigenvalues, the corresponding eigenvectors of $J_{f^{-1}}$ across the samples could be very different, so is this averaging meaningful?

---

> > > ### Author Response · Authors · 2022-08-09
> > > **Meaningful averaging**
> > >
> > > Indeed, in general, the eigenvectors across the samples will be very different. However, we are not interested in the eigenvectors as such but rather in the magnitudes of their eigenvalues. According to our theory, the largest eigen/singular values will correspond to singular vectors pointing in the off-manifold directions. For instance, we can think about a circle embedded in $\mathbb{R}^{2}$ inflated by adding some isotropic Gaussian noise with small variance $s^{2}$. For each sample, the largest eigen/singular value will correspond to the singular vector pointing towards the radial direction (i.e. the directions normal to the circle). Across the different samples, all those largest eigen/singular values will have the same magnitudes despite being associated with different normal directions.
> > >
> > > More generally, we are only interested in the cut-off separating eigen/singular values corresponding to on- and off-manifold directions. Therefore, the averaging reduces the noise introduced by limited sample size or unexact learning of $q_{\sigma^{2}}$.
> > >
> > > Below we show the results on the toy-examples if we do not average over the eigen/singular values but over the local estimation of $\hat{d}(k)$ for each sample $x_{k}$, i.e. $\hat{d}=K^{-1}\sum_{k=1}^K\hat{d}(k)$. This estimate is still very good, though with greater variability. For easier comparison, we added the result when averaging over the eigen/singular values in the last column.
> > >
> > > |Distibution  				| D | ID | ID-NF| paper |
> > > |-------------|---|----|-------|----|
> > > |mixture on sphere| 		3 | 2| 2.0 $\pm$ 0.13| 2.01 |
> > > |correlated on sphere|		 3| 2| 2.08 $\pm$ 0.05| 2.04 |
> > > |mixture on torus| 			3| 2| 2.02 $\pm$ 0.11| 2.02 |
> > > |correlated on torus| 		3| 2| 2.0 $\pm$ 0.19| 2.02 |
> > > |correlated on hyperboloid|	 3| 2| 2.03 $\pm$ 0.02| 2.02|
> > > |unimodal on hyperboloid|	 3| 2| 2.03 $\pm$ 0.04| 2.01|
> > > |exponential on thin spiral| 	 2| 1| 1.0 $\pm$ 0.0| 1 |
> > > |mixture on swiss roll| 		3| 2| 2.05 $\pm$ 0.3| 2.02 |
> > > |correlated on swiss roll| 		3| 2| 1.96$\pm$ 0.46| 2.02|
> > > |mixture on stiefel| 			3| 1| 1.31 $\pm$ 0.46| 1.07|

---

> > > > ### Comment · Reviewer_aewG · 2022-08-10
> > > > **Update**
> > > >
> > > > Thanks for your detailed response. My concerns are well resolved.

---

### Official Review · Reviewer_WmkV · 2022-07-11

**Rating:** 6
**Confidence:** 3
**Soundness:** 3 good
**Presentation:** 3 good
**Contribution:** 3 good

**Summary:**

This paper introduces a novel NF-based algorithm to estimate intrinsic dimension. The algorithm is based on the observation that large variability is connected to small eigenvalues of the Jacobian, and that the variance of intrinsic noise can affect where the eigenvalue changes significantly. To estimate intrinsic dimension, this paper fits the Jacobian, and further uses two sigmoidal functions to fit the thresholding function that should have a plateau. Empirically, this paper compares the algorithm with prior methods on several experiments where the true intrinsic dimension is known.

**Questions:**

- What is the theoretical guarantee of the proposed algorithm? This question can include a lot of things, such as average-case guarantee or worst-case guarantee, bounds on errors or failure probabilities, etc. Do you have answers to at least one of these questions?

- What is the complexity of the proposed algorithm?

- Do you have some assumption on the NF model, as different NF models behave very differently? Are there differences if you test different NF models in your experiments?

- How did you choose $N$ and noise magnitudes $\bar{\sigma}_n^2$ for $n=1,\cdots,N$? Is there a theoretically optimal schedule? What is the affect of changing this schedule, either theoretically or empirically?

- Does your algorithm lead to successful, practical applications that use intrinsic dimensions? For example, after running the proposed algorithm, you can find the correct latent dimension for a GAN model, and it outperforms other latent dimensions.

**Limitations:**

The authors indicate there is no negative social impact as this is a theoretical paper, which I agree with.

-----------------------------

After rebuttal, I decide to increase my score to 6.

**Strengths And Weaknesses:**

I am not an expert in the area of estimating intrinsic dimension, so I would like to leave the evaluation of originality and significance to other reviewers. I think this paper is clearly written, and the intuition is easy to follow.

The methodology of this paper is one major strength. Using NF to estimate intrinsic dimension seems to be an interesting idea to me. The Jacobian matrix and its eigenvalues offer a nice way to investigate the sensitivity (variability) for each direction. In short, the entire section 4 makes a lot of sense to me. I especially like the idea of adding different magnitudes of Gaussian noises and training separate NFs.

Empirically, the proposed algorithm can indeed estimate intrinsic dimension very accurately and outperform prior methods, which is another strength.

Despite the above strengths, I think this paper lacks some key points. On the theory side, the algorithm does not come with theoretical guarantee, for example, approximation properties or (probabilistic) error bounds. Also, the choice of NF $f$ and noise magnitudes $\bar{\sigma}_n^2$ can potentially affect results by a lot, but there is neither theoretical nor empirical discussion on these. Finally, there is no such experiment that the proposed algorithm finds the intrinsic dimension in a practical problem (e.g. of a real dataset) and lead to successful application (e.g. help fixing the latent dimension of a generative model).

---

> ### Author Response · Authors · 2022-07-31
> **Answer to WmkV**
>
> 1. We refer to the "Theoretical additions" answer to provide some theoretical backing for Observations 1 and 2 on which the Algorithm is based. We thank the reviewer for sharing these questions. We would need some more time to address them precisely.
>
> 2. The complexity of the algorithm depends on a) the complexity $C$ to train an NF b) the number $N$ of NFs used for training c) calculating the Jacobian of $K$ samples, and d) calculating the eigen/singular value decomposition of the Jacobian. Therefore, the training complexity (a) and b)) is given by $\mathcal{O} (N\cdot C)$. The complexity of c) depends on the operational complexity of the NF, see [1]. This depends on the number of operations needed for one forward pass of the NF. Denoting this complexity with $C_{NF}$, the total complexity of c) is $\mathcal{O}(K\cdot C_{NF})$. The complexity of d) is $\mathcal{O}(D^3)$. Therefore, in total, the complexity of the algorithm is $\mathcal{O}(N\cdot C)+\mathcal{O}(K\cdot C_{NF}) + \mathcal{O}(D^3) $ where the first term refers to the training- and the remaining terms to the evaluation complexity.
>
> 3. There is no explicit assumption on the NF model. For the non-image datasets, we used a Block Neural Autoregressive Flow (BNAF). For the images, we used neural splines in combination with coupling layers. We are aware of the existence of inductive biases for the affine-linear flow, see [2]. We thank the reviewer for raising this point and will conduct some additional simulations to examine the sensitivity to the particular flow model. As a preliminary result, we failed to learn the uniform distribution on the sphere $S(200)$ using a shallow NF based on affine-linear coupling layers. Thus, the expressiveness of the flow seems to be important.
>
> 4. We did not follow a particular schedule for choosing $N$ and the inflation noises. We simply set $\bar{\sigma_{1}^{2}}$ to be very small (in the order of $10^{-9}$) and $\bar{\sigma_{N}^{2}}$ to be very large (in the order of 10 for the toy examples). The remaining $\bar{\sigma_{i}^{2}}$ we spread equidistantly.
>
>    For bounded data, $N=3$ is a good choice as discussed in the general answer. In this case, we estimated the critical value for which the singular values should decay heuristically.
>
>    For unbounded data, $\bar{\sigma_{N}^{2}}$ must be sufficiently large to see a decay of the singular values corresponding to directions of large variability such that $F(\alpha)$ can be well estimated (see equation (10)). This critical value can be related to the so-called reach number of a manifold:
>    Informally, this number refers to the maximum distance to the manifold such that a unique projection onto the manifold exists, see [5] or [6]. Thus, for noise levels beyond this reach number, the manifold is garbled too much and directions of small or large variability cannot be differentiated. Hence, the reach number could serve as a critical value for $\sigma^{2}$ for unbounded data. Once such a critical value is known, again only a few NFs are needed to estimate the ID. There exist methods to estimate this reach number, see e.g. [7].
>
> 5. We did not conduct such experiments. We thank the reviewer for this suggestion. As mentioned in the introduction, two recently introduced latent variable models based on NFs for manifold valued data do require knowing the exact dimensionality of the manifold, see [3] and [4]. Both methods perform relatively poorly on the CelebA-HQ dataset for which $d$ is unknown. As an estimate, they set $d=512$. It would be interesting to see those methods' performance using our algorithm's output as latent dimensionality. We will conduct this experiment and update you on the outcome - hopefully before the discussion period ends.
>
> References:
>
> [1] Margossian, Charles C. "A review of automatic differentiation and its efficient implementation." Wiley interdisciplinary reviews: data mining and knowledge discovery 9.4 (2019): e1305.Charles C. Margossian. A Review of Automatic Differentiation and its Efficient Implementation. arXiv:1811.05031, 2019.
>
> [2] Kirichenko, Polina, Pavel Izmailov, and Andrew G. Wilson. "Why normalizing flows fail to detect out-of-distribution data." Advances in neural information processing systems 33 (2020): 20578-20589
>
> [3] Johann Brehmer and Kyle Cranmer. Flows for simultaneous manifold learning and density estimation. Advances in Neural Information Processing Systems, 33, 2020.
>
> [4] Christian Horvat and Jean-Pascal Pfister. Denoising Normalizing Flow. In Advances in Neural Information Processing Systems, volume 34, pages 9099–9111. Curran Associates, Inc., 2021.
>
> [5] Christian Horvat and Jean-Pascal Pfister. Density estimation on low-dimensional manifolds: an329
> inflation-deflation approach. arXiv:2105.12152
>
> [6] Clement Berenfeld and Marc Hoffmann. Density estimation on an unknown submanifold. arXiv:1910.08477
>
> [7] Aamari, Eddie, et al. "Estimating the reach of a manifold." Electronic journal of statistics 13.1 (2019): 1359-1399.

---

> > ### Comment · Reviewer_WmkV · 2022-08-04
> > **Discussion**
> >
> > Thank you for your response -- it answered some of my questions. As for my question 1 regarding theoretical guarantee, observations 1 and 2 totally make sense to me (at least from a high level). I would like to see if there is any analysis on algorithm 4 as an estimation method. For example, can you guarantee that the estimated $\hat{d}$ is close to the real intrinsic dimension $d$, at least under some assumptions (e.g. you can assume the NFs are very good)? Is it an unbiased estimator? I think such analysis is very necessary for an estimator to be theoretically sound, and therefore I would like to see results on this.

---

> > > ### Author Response · Authors · 2022-08-05
> > > **Analysis on algorithm**
> > >
> > > We thank the reviewer for insisting on a theoretical analysis of the algorithm. We will give informal proof for its convergence in the following.
> > >
> > > Let $K=1$, i.e. we want to estimate $d$ locally for a given sample. We already commented on the choice of $N$ and $\bar{\sigma_{n}}^{2}$. We assume that the noise magnitudes are chosen such that all eigenvalues decay, and that all inflated densities $q_{\bar{\sigma_{n}}^{2}}$ are learned exactly. Besides these necessities, our estimate for $\hat{d}$ crucially depends on
> > >
> > > 1. the ability to estimate the eigenvalue trajectories $\lambda(\sigma^2)$.
> > > 2. the manifestation of a "plateau" on the $F(\alpha)$ plot with height corresponding to the manifold normal space dimension
> > > 3. the finding of this plateau by fitting a sum of two sigmoidal functions.
> > >
> > > *On 1.* It is easy to see that our model $\hat{\lambda}(\sigma^{2}; \alpha^{2}, \beta) = \beta / \sqrt(\alpha^{2}+\sigma^{2}), \beta>0,$  for estimating these trajectories is convex in its parameters $\alpha^{2},\beta$. Therefore, these parameters can be learned exactly.
> > >
> > > *On 2.* As discussed in the additional "Theoretical additions" answer, a flow $f^{-1}$ transforms a local neighborhood of a point $x$ to a Gaussian with covariance given by the Gram matrix of the flows Jacobian. More precisely, points generated by adding Gaussian noise to $x$ with variance $s^{2}$ will be mapped onto a Gaussian with covariance $J_{f^{-1}}^{T} J_{f^{-1}}$ if $s^{2}$ goes to $0$.
> > >
> > > In the paper, we discussed this Gaussian case in detail and argued why directions of large variability will decay much later than directions of small variability manifesting in a plateau. Thus, since we have learned the inflated distributions $q_{\bar{\sigma_{n}}^{2}}$ exactly, the plateau in the $F(\alpha)$ plot must manifest. Since $F(\alpha)$ is simply a sum of step functions where the onset of a step represents the start of decay of an eigenvalue, the height of this plateau is exactly the number of normal space directions/ directions of small variability.
> > >
> > > However, it is important to note that for rigorous proof, we would need to make some assumptions about the on-manifold density. For example, think of a Gaussian on a 2d plane embedded in $\mathbb{R}^{3}$. If the variance of this Gaussian is much higher in one direction than in the orthogonal direction, the plateau could be at height $2$ rather than $1$. Therefore, intuitively, the differences in the variances of two neighboring eigenvalues corresponding to on-manifold directions must be smaller than the smallest eigenvalue corresponding to on-manifold directions.
> > >
> > > *On 3.* Once we know that the plateau corresponds to the right height, can we find it? In the paper, we proposed a sum of 2 sigmoidal $sig(\alpha;a,b,c)= \frac{a}{1+\exp(-b(\alpha-c))}$. If we use $D$ instead of 2 sigmoidal, we need to fix $D$ sigmoidal to $D$ step functions. One can approximate a step function with a sigmoidal arbitrary well, see [1]. As $F(\alpha)$ is a sum of step functions, we can learn it arbitrarily well. Once we have learned the parameters $a,b,c$ of those sigmoidal, we can read out the plateau using the differences of the $c$ parameters (which are the symmetry points of the sigmoidals and thus encode the jump onsets).
> > >
> > > What if we use only 2 instead of $D$ as proposed in the paper? We can show that if there is a plateau, using two instead of 1 sigmoidal lead to a better fit. Thus, the heights of the sigmoidals $a_1$ and $a_2$ are positive. Note that the cost is convex in $a>0$ (the step size), hence what is left to show is that a model with $a_1 \neq D-d$ will lead to a greater cost. The latter is easy to see if we assume that all the step onsets prior to the plateau are very close together or, at least, equidistant. The former is true if the intrinsic noise is the same in all directions.
> > >
> > >
> > > Reference:
> > > 1. Kyurkchiev, Nikolay, and Svetoslav Markov. "Sigmoid functions: some approximation and modeling aspects." LAP LAMBERT Academic Publishing, Saarbrucken 4 (2015).

---

> > > > ### Comment · Reviewer_WmkV · 2022-08-06
> > > > **Discuss**
> > > >
> > > > Thank you for your reply. The analysis makes sense to me. It would be nice to have a formal proof in the paper. I will take it into consideration when I make the final decision.

---

> > ### Author Response · Authors · 2022-08-09
> > **Update to point 5**
> >
> > As promised, we conducted such an experiment. Please see our answer "Experiments on improving latent variable models" in the discussion with Reviewer ozqo.

---

### Official Review · Reviewer_ozqo · 2022-07-11

**Rating:** 6
**Confidence:** 3
**Soundness:** 3 good
**Presentation:** 2 fair
**Contribution:** 3 good

**Summary:**

This paper discusses the use of normalizing flows for estimating the intrinsic dimensionality (d) of the data sub-manifold embedded in high-dimensional ambient space ($\mathbb{R}^D, d < D$). The authors claim the proposed method based on flows is scalable and accurate compared to some standard techniques, e.g., nearest neighbor searches. The proposed method is based on a PCA-like rule, i.e., an eigendecomposition of a covariance transformation, by approximating the non-linear flow as a linear map with a Jacobian. The authors propose two techniques that can estimate whether a certain eigenvector is an on-manifold direction or not: the first is simply checking out the magnitude of the corresponding eigenvalue and the latter is judging the sensitivity of eigenvalues by injecting some noises. The authors validate their proposed method experimentally, compared with the nearest neighbor and another flow-based approach (LIDL) for some simple examples and StyleGAN2 manifolds.

**Questions:**

[Major comments]

1. It seems the estimated intrinsic dimensionality is highly dependent on the manifold approximation performance of flows (i.e., the generative power of flows). Thus, it is not entirely clear whether the proposed method is suitable for extremely-complicated high-resolution datasets. Note that the method requires a down-sampling of StyleGAN-generated images (from $(1024, 1024, 3)$ to $(64, 64, 3)$), and fails to correctly estimate the dimensionality of such a generated manifold ($d=64$ but $\hat{d}=53$).

2. While the authors state the proposed method is scalable, there is no theoretical analysis as well as experimental demonstration regarding it. Because the proposed method should train a set of $N$ flows, it is not entirely sure it is indeed cheap. Please elaborate more.

3. What would be the killer application of the intrinsic dimensionality estimation? Can it be used to, e.g., improve generative performance, find disentangled representations, or detect anomaly samples correctly? It will be nice if the authors state some possible applications of the proposed method by demonstrating a proof-of-concept experiment regarding it.

[Minor comments]
1. It is not clear why $\beta_i$ should be used.

2. In the pedagogical example 4.1, it is not clear why the eigenvalues of on-manifold directions are different (see Fig. 1 (a)) while the underlying sub-manifold is a 2-sphere.

3. Observation 2 is not clear because (9) does not contain the injected noise $\sigma$ explicitly (while one may infer $\lambda^{(i)} = 1/ \sqrt{\sigma_{0}^{2} + \sigma_{i}^{2} + \sigma^{2}}$).

4. In line 159 of page 4, the authors mention $\beta_i / \alpha_i = \lambda^{(i)}$ must be much larger for large variability directions, while in Observation 1 they mention large eigenvalues correspond to directions of small variability.



**Limitations:**

The authors state the limitation of the proposed method in the Discussion section, e.g., the possible computational cost (which is consistent with this reviewer’s major comment 2) and the theoretical assumption on the boundness of the data manifold. In my opinion, the former is not fully addressed, thus should be elaborated more for a convincing presentation of the manuscript. The latter seems to be addressed well by proposing some practical solution to deal with the bounded data, e.g., $0.68 * x_{max}$ for image data.

**Strengths And Weaknesses:**

[Strengths]

The main idea of this paper is simple and principled. It can be derived from the fundamental principle of linear algebra, thus easy-to-follow. The paper is generally well-written, while there are some confusing points (see minor comments). The proposed method outperforms other competitors including LIDL (which might be the SOTA so far), for some synthetic examples.


[Weaknesses]

I am not fully convinced with the proposed method, especially whether it can be used for a more complicated manifold regarding both in terms of accuracy and scalability (see major comments 1 and 2). In addition, I cannot find strong applications of the proposed method (see major comment 3).

---

> ### Author Response · Authors · 2022-07-31
> **Answer to ozqo**
>
> 1. Our method depends on the flow's ability to transform samples from an arbitrary density $p(x)$ to samples from standard Gaussian. Note that this is related  though different to the flow's ability to estimate $p(x)$ exactly. Indeed, the experiments on the high-dimensional synthetic datasets show that we can estimate $d$ well, even for $10^{3}$ samples - as opposed to the LIDL method which directly depends on the flow's ability to learn $p(x)$ exactly.
>
>    The reviewer claims that our method requires a down-sampling of StyleGan-generated images from $(1024,1024,3)$ to $(64,64,3)$. This is not true. We downsample to accelerate the training time as training an NF on images with a resolution of $(1024,1024,3)$ would require many days (see e.g. [1] where an NF was trained on $(256,256,3)$-resolution images). Therefore, we don't strictly require this downsampling but did it for the sake of training efficiency.
>
>    Note, that our estimate for the $d=64$ dimensional manifold is correct now (see general answer).
>
> 2. The scalability refers to a) the number of data points and b) the embedding dimension. Since our method requires training NFs, which do scale to large datasets and large dimensions, our method does as well. We we experimentally confirmed the latter through estimating the ID for high-dimensional datasets (spheres and images). Though, we agree that we did not address the computational complexity of our method accurately. As an orientation, training an NF on any non-image datasets was very fast - only a few minutes on a GPU. The entire experiment on the sphere $S(200)$ embedded in $D=400$ experiment takes only 10 minutes. For a more thorough answer to the complexity, we refer to our answer to the second question of Reviewer WmkV.
>
> 3. Estimating the ID of a data-manifold is relevant for many fields, e.g. see [2] and the references therein. We assume that the reviewer refers to a killer application in the field of Machine Learning. Indeed, one interesting application would be to improve the generative performance of latent variable models. As an example, recently developed latent variable models based on NFs  do need to know the exact number of latent variables, see [3] and [4]. Hence, our method could allow these methods to be applied on real-world datasets where the ID is unknown. As mentioned in the answer the last question of reviewer WmkV, we will conduct this experiment and keep you updated regarding the outcome, if time allows.
>
>  Another very interesting application would be out-of-distribution detection. How will the eigen/singular values of out-of-distribution samples evolve? Intuitively, the ID for those samples should be greater. We will test this for the StyleGan image manifolds and keep you updated.
>
> References:
>
> [1] Diederik P Kingma and Prafulla Dhariwal. Glow: Generative flow with invertible 1x1 convolutions. arXiv:1807.03039, 2018.
>
> [2] Facco, Elena, et al. "Estimating the intrinsic dimension of datasets by a minimal neighborhood information." Scientific reports 7.1 (2017): 1-8.
>
> [3] Johann Brehmer and Kyle Cranmer. Flows for simultaneous manifold learning and density estimation. Advances in Neural Information Processing Systems, 33, 2020.
>
> [4] Christian Horvat and Jean-Pascal Pfister. Denoising Normalizing Flow. In Advances in Neural Information Processing Systems, volume 34, pages 9099–9111. Curran Associates, Inc., 2021.

---

> > ### Author Response · Authors · 2022-08-02
> > **Update: Out-of-distribution experiment succesfull**
> >
> > We estimated the ID of 10 StyleGan 64d generated images using the trained NFs on StyleGan 2d images and could successfully detect them as OOD. The ID for those 10 images was significantly greater than 2 - in the order of 50. This was to be expected as those images live in space normal to the manifold which is higher dimensional. We think this is a very interesting result and could be a potential killer application. We will include it in the revised version.

---

> > > ### Comment · Reviewer_ozqo · 2022-08-08
> > > **Reply to the authors**
> > >
> > > I appreciate the authors’ thorough response.
> > >
> > > 1. Thank you for clarifying the StyleGAN experiment. I am generally satisfied with the authors' explanation and new result that estimate the manifold dimension (64) correctly.
> > >
> > > 2. I read the response to Reviewer WmkV as well as this rebuttal. I am still not sure whether the cost of the proposed method is indeed cheap. However, I think  the paper’s contribution outweighs the concern regarding the scalability issue.
> > >
> > > 3. I am happy to see the additional study regarding the out-of-distribution (OoD) detection, which is very intereseting and seems to be potentially useful.
> > >
> > > Based on the authors' rebuttal, I increased my score as 6. I would like to recommend authors to adding the newly conducted experimental results as well as clarifying minor things in my review in their final version of the paper (if the paper is accepted).
> > >
> > > Minor comment: Are the authors willing to release their code to reproduce the results?

---

> > > > ### Author Response · Authors · 2022-08-09
> > > > **Experiments on improving latent variable models**
> > > >
> > > > We thank the reviewer for the response. Regarding the code: yes, we are planning to release our code upon acceptance.
> > > >
> > > >
> > > > *Updates regarding killer application*:
> > > >
> > > > As promised, we conducted an experiment to see if our method can be used to improve the performance of latent variable models. In particular, we focused on models based on NFs such as the manifold flow ($\mathcal{M}-$flow) introduced in [1] and the Denoising Normalizing Flow (DNF) introduced in [2]. Both methods set the latent dimension to $512$ for the CelebA-HQ (downsampled to a resolution of 64x64x3). However, this number was chosen somewhat arbitrarily as the true ID (if exists) is unknown for this real-world dataset. We used our method to estimate the ID and got $\hat{d}=130$. We trained an DNF with the same architecture as in the original paper using  a latent dimensionality of $130$ instead of $512$. After 300 epochs, we get an FID score of 36.92. The original DNF has an FID score of 34.14, and the $\mathcal{M}-$flow of 38.07 - after 500 epochs of training. Thus, we can get very similar results in terms of generative power (measured by the FID score) with only $130$ latent dimensions instead of $512$. This is a very interesting result showing how our method can be used to apply models such as [1] and [2] on real-world datasets.
> > > >
> > > > References:
> > > >
> > > > [1] Johann Brehmer and Kyle Cranmer. Flows for simultaneous manifold learning and density estimation. Advances in Neural Information Processing Systems, 33, 2020.
> > > >
> > > > [2] Christian Horvat and Jean-Pascal Pfister. Denoising Normalizing Flow. In Advances in Neural Information Processing Systems, volume 34, pages 9099–9111. Curran Associates, Inc., 2021.

---

### Official Review · Reviewer_Vtze · 2022-07-11

**Rating:** 8
**Confidence:** 3
**Soundness:** 3 good
**Presentation:** 3 good
**Contribution:** 4 excellent

**Summary:**

This paper focuses on an interesting and important problem in machine learning: For a given set of data around a low-dimensional manifold embedded in a high-dimensional R^D, how can we estimate the intrinsic dimension (ID) of the manifold. The traditional estimator based on the nearest neighbor statistics cannot be applied to large datasets due to the complexity and suffers from the curse of dimension. In this paper, a normalizing flow (NF) based method is proposed, which solves the problem by using the following steps: 1) Add Gaussian noise with different variances to the dataset. 2) Estimate the density functions for noisy data with different variances by a set of NFs. 3) Estimate the ID from the eigenvalues of Jacobian matrices of tranformations defined by NFs.

**Questions:**

On the justification:

1) The main "theoretical" basis of the method is Observation 1, but it is just an "observation" instead of a theorem (even an "informal" theorem). So there's a lot of ambiguity here. What are definitions of variability and direction of variability? What does small/large variability mean? How can we quantify the variability? As an example, does the variability of a standard Gaussian distribution have the same magnitude for all directions? If so, how about the uniform distribution or other isotropic probability distributions?

2) In addition, does Eq. (8) mean that the "variability" is a constant for any position for a standard Gaussian distribution? Can we choose the target distribution of f^{-1}(x) as uniform distribution or other isotropic probability distributions?

3) The similar question for Observation 2: Why can the conclusion on the Gaussian distribution be extended to a general noisy data-manifold?

As a summary, I believe the observations are reasonable, and the method developed based on the observations really works. But more thorough analysis and explanations are required. I hope to see two theorems (even informal theorems based on some specific assumptions) in the revision.

Some other comments:

4) In Lines 43-50, a very interesting problem is proposed: The dataset may be on a union of manifolds, and the local ID is more useful than the global one in this case. But all experiments in this paper are on estimation of global ID. Can the NF method be applied to estimation of local ID? Is there any difficulty?

5) It seems that the intrinsic noise is assumed to be Gaussian and spatial homogeneous. Will the performance of the algorithm be greatly affected if the assumption does not hold?

6) Besides the "dimension number", people are also interested in the component related to each dimension. Can we get, for example, the projection of data on the dimensions, from eigenvectors of Jacobian matrices of NFs?

7) The paper is highly related to the SoftFlow proposed in [22]. In SoftFlow, one NF is trained for different noise magnitudes with the noise magnitude being a parameter of the NF. This technique may be also helpful for this paper since SoftFlow can easily obtain the structure of the data independent of noise.

**Limitations:**

Authors have adequately addressed the limitations and potential negative societal impact of their work.

**Strengths And Weaknesses:**

Strengths:

Authors use the structure and property of NFs in a very clever way and show that we can obtain the information on ID from eigenvalues of Jacobian matrices of NFs. As far as I know, the eigenvalues of the Jacobian matrices of are rarely noticed in the research on NFs. In addition, the method is technically sound, and experiments also demonstrate the effectiveness of the proposed method.

Weaknesses:

The main weakness of the paper comes from that the justification of the proposed method is not well formulated. See below for details.

---

> ### Author Response · Authors · 2022-07-31
> **Answer to Vtze**
>
> 1. We agree that our Observations induce some ambiguity since we do not formally define terms such as 'direction of variability'. When we talk about variability, we have the following geometrical picture in mind.
>
>    Consider equation (9). The flows Jacobian linearly maps a local neighborhood of $x*$ to a local neighborhood of $f^{-1}(x*)$ in the latent space. Let this local neighborhood be a ball with radius $\sigma^{2}$. This linear mapping will transform the ball into an ellipsoid with the change in volume proportional to the Jacobian determinant. Now, the singular vectors of the Jacobian will map to the axis of the ellipsoid. These singular vectors are the directions of variability, and the singular values quantify the variability. Note that in the case of Gaussian distributed data, these directions refer to the variability in the data itself (similar to the principal components in PCA).
>
> *Edit: Since the covariance matrix is symmetric with non-negative eigenvalues, the singular- and eigenvalues coincide.*
>
> 2. Therefore, indeed the variability is a constant at any given position for a flow mapping a standard Gaussian to a standard Gaussian distribution. However, if we were to choose another target distribution (uniform or another isotropic distribution), it would not be constant anymore. We are currently studying the effect of the target distribution on our method - it is a work in progress.
>
> 3. Please see our separate answer providing more theoretical backing for our theory.
>
> 4. Indeed, our method can be used as a local estimator, see Remark 1 (i). No additional difficulty applies. As experimental evidence, we applied our method to the Lolipop dataset proposed by the LIDL paper [1]. As LIDL, we accurately estimate the ID locally (1 for the "stick" region and 2 for the "disk" region). We will include this experiment in the appendix.
>
> 5. As long as an isotropic noise is used, no performance drop is to be expected. As  experimental evidence, we repeated the experiments on $S(200)$ embedded in $\mathbb{R}^{400}$ with uniform instead of Gaussian noise using $10^{3}$ samples. We get very similar results as with Gaussian noise - the output of our algorithm is $200.95$.
>
> 6. This is a very interesting thought. In principle, the eigen- or singular vectors can be used to project the data into a lower-dimensional space - similar to PCA. However, for curved manifolds, at any given point, these vectors will span a different subspace. It is therefore not clear how these projections will relate to each other, and if they would be useful as a low-dimensional representation. However, we could imagine some iterative scheme, moving with small steps along the manifold and correcting the displacement of eigen/singular directions. In any case, how the eigen/singular directions can be further exploited is an interesting research direction to pursue.
>
> 7. This is an interesting idea to reduce the NF's complexity. We will leave this for future work.

---

> > ### Author Response · Authors · 2022-08-03
> > **Update regarding point 6**
> >
> > We were curious if the singular directions could indeed be used to find low-dimensional representations. For that, we trained a standard flow on uniforms samples on a swiss roll using a) a standard flow b) a flow with uniform target distribution.
> >
> > In both cases, after projecting the samples onto the plane spanned by the singular vectors corresponding to the two largest singular values, we get a 2-dimensional embedding able to nicely separate different parts of the swiss-roll. Although the embedding is not exactly an "unrolling" of the swiss roll, it is still a meaningful/useful  2d representation.  Using the uniform as a target distribution leads visually to a slightly better separation (as expected). This is a nice first result calling for more research. We thank the reviewer for raising this point.

---

### Author Response · Authors · 2022-07-31
**General answer**

We sincerely thank all the reviewers for taking the time to have read our manuscript attentively. We are especially delighted by the many interesting questions and comments raised. We will address all of them separately for the sake of clarity and we are looking forward to the reviewer-author discussion period in case further clarification is needed.

From a high-level point of view, all the reviewers agree that our method would benefit from a greater theoretical foundation of Observations 1 and 2. Especially given the fact that it works so well even for non-Gaussian data. We will address these issues in a separate answer entitled "Theoretical additions" by providing some theoretical evidence why the general case can be traced back to the Gaussian case.

We would like to inform the reviewers that, since our submission, we were able on decrease the complexity of our proposed method for image datasets. In the submitted manuscript, we used $N=30$ Normalizing Flows (NFs) for estimating the intrinsic dimensionality (ID) of the StyleGan image manifolds. However, it is enough to only train 3 NFs: one with a very small $\sigma^{2}$, one with $\sigma^{2}=(0.68\cdot 255)^2$ (which is based on the calculations presented in Section 4.2), and one with a very large $\sigma^{2}$ as we are only interested in the onsets of decay after this critical value $\sigma^{2}=(0.68\cdot 255)^2$.

Using these 3 NFs, we are excited to report that our estimate for the StyleGan d=64 is exact now, i.e. we get $\hat{d}=64$. On the one hand, training more NFs introduces more variability in the $\lambda$ vs. $\sigma^{2}$ curves used to estimate the ID. This variability leads to more variability in the parameter estimates of $\beta$ and $\alpha$ defining $\hat{\lambda}$. On the other hand, we increased the number of epochs used to train the NFs from 50 to 200. These could be some explanations for why our previous estimate was not exact.

Finally, we would like the reviewers to (re-)consider the seminal character of this work. To the best of our knowledge, our method is the first one able to estimate the ID of such high-dimensional image manifolds consistently. We believe that this work will have a great contribution to a better understanding of how such datasets are structured - especially from the perspective of the manifold hypothesis.

---

> ### Author Response · Authors · 2022-07-31
> **Theoretical additions**
>
> Observation 1 states that the magnitudes of eigenvalues/singular values determine direction of variability. Observation 2 describes how these values change depending on the inflation noise. In the following, we will give some theoretical explanations why theses observation generalize to non-Gaussian data.
>
> *Tracing back the Gaussian from the non-Gaussian case*:
>
> Consider a flow $f$ mapping a standard Gaussian sample $z$ to data $x=f(z)$ distributed according to $p(x)$. Lemma 1 in [2] "says that points generated by a flow in small region around a fixed point $x$ will be approximately distributed to Gaussian with mean $x$ and covariance " $J_{f}J_{f}^{T}$. The the same is true for the inverse direction as well with covariance $J_{f^{-1}}J_{f^{-1}}^{T}$. With other words, a flow transforms not only globally a distribution into a Gaussian, but also locally. Thus, the singular vectors (which are the eigenvectors of the Gram matrix $J_{f^{-1}}J_{f^{-1}}^{T}$) show in directions of variability in the data (see our answer to the first question of Reviewer Vtze), and Observations 1 and 2 are valid for non-Gaussian distributed data as well.
>
> *Theoretical addition to Observation 1:*
>
> Another more geometrical explanation for Observation 1 involves the interpretation of the Jacobian determinant as volume changing factor. If no noise is added, the singular values in off-manifold directions must theoretically be $\infty$ while the singular values in on-manifold directions must be finite. Now, adding isotropic noise will not change the orientation of singular directions. Therefore, the $d$ lowest singular values will still be in the manifold direction whereas the remaining will be in the off-manifold direction since the singular vectors are orthonormal.
>
> *Theoretical addition to Observation 2:*
>
> Let $\mathcal{M}$ be a d-dimensional manifold $\mathcal{M}$ embedded in $\mathbb{R}^{D}$. We denote a density on $\mathcal{M}$ by $p(x)$ and the corresponding latent distribution by $\pi(u)$. We assume that the support of the generating latent distribution is diffeomorphic to $\mathbb{R}^{d}$ such that we can w.l.o.g. set $\pi(u)$ to be standard Gaussian. We denote the corresponding injective mapping from latent to data space by $f(u)$.
> Now, we inflate the data manifold with Gaussian noise of magnitude $\sigma^{2}$ but with the strong assumption that the noise is only added to the manifold's normal space. Thus, the new generating mapping is given by $\tilde{f}(u,v) = f(u) + A_{u} v$ where $u \sim \pi(u)$ and the columns of $A_{u}$ span the normal space in $x=f(u)$. In the case of standard NFs, we sample $v$ from a standard Gaussian. Then, the  singular values of $A_{u}$ must depend on $\sigma^{2}$, but, crucially, $f(u)$ (and therefore the Jacobian $J_{f}$) is independent of $\sigma^{2}$. Therefore, only singular values corresponding to directions of small variability (normal space directions) will change. This is the gist of Observation 2.
>
> Note that for the special case of $D\gg d$, standard Gaussian noise is an excellent approximation for a Gaussian restricted to the normal space as, intuitively, all directions are essentially normal. This was proven in [1] and explains why Observation 2 can be used for the StyleGan datasets where $D\gg d$.
>
> References:
>
> [1] Christian Horvat and Jean-Pascal Pfister. Density estimation on low-dimensional manifolds: an329
> inflation-deflation approach. arXiv:2105.12152
>
> [2] Cunningham, Edmond, Adam Cobb, and Susmit Jha. "Principal manifold flows." arXiv preprint arXiv:2202.07037 (2022).

---

### Author Response · Authors · 2022-08-09
**Summary discussion**

We thank all the reviewers for this very fruitful discussion period. It will help us to improve our manuscript substantially. In particular, we commit to the following changes for the final version:

+ Update the result on the StyleGan 64d image manifold using only $N=3$ NFs ( see our *"General answer"*)
+ We will clarify what we mean by directions of small and large variability - in line with our *"Answer to Vtze"*, point 1.)
+ we will add 3 formal theorems/lemma:
  1. a flow maps not only globally a distribution to a Gaussian but also locally (in line with our *"Theoretical additions"* answer, *"Tracing back the Gaussian from the non-Gaussian case"*)
  2. The convergence of our algorithm under some specific assumptions (see our *"Analysis on algorithm"* answer)
  3.  For the special case if noise is only added to the manifold's normal space, only the singular values in off-manifold directions will change (in line with our *"Theoretical additions"* answer, *"Theoretical addition to Observation 2:"*)
+ We will add additional experiments/results:
  1. our method applied to the Lolipop dataset (see our *"Answer to Vtze"*, point 4.)
  2. Using uniform instead of Gaussian noise does not change the result  (see our *"Answer to Vtze"*, point 5.)
  3. An extended OOD experiment based on our answer *"Update: Out-of-distribution experiment succesfull"*
  4. The latent variable model experiment as described in our answer *"Experiments on improving latent variable models"*
  5. A complexity analysis of the algorithm, see our *"Answer to WmkV"*, point 2.
  6. The sensitivity to the expressiveness of the flow, see our *"Answer to WmkV"*, point 3.
+ we will make our code publicly available

---

### Meta-Review · Area_Chair_yP4S · 2022-08-25

**Recommendation:** Accept
**Confidence:** Certain

**Metareview:**

This paper discusses the use of normalizing flows for estimating the intrinsic dimensionality of the data sub-manifold embedded in high-dimensional ambient space. The idea of using NFs to estimate the intrinsic dimensionality via analyzing the eigenvalues of the Jacobian matrices is novel. The method is technically sound, and the paper presents detailed experiments on pedagogical examples, low-dimensional synthetic examples, and high-dimensional images generated by StyleGAN, which show the effects of this method.


**Award:**

No

---

### Decision · Program_Chairs · 2022-09-14

Accept